# Emergent XY* transition driven by symmetry fractionalization and anyon condensation

Michael Schuler[1*], Louis-Paul Henry[2], Yuan-Ming Lu[3] and Andreas M. Läuchli[4,5]

1 Institut für Theoretische Physik, Universität Innsbruck, A-6020 Innsbruck, Austria
2 Pasqal, 2 avenue Augustin Fresnel, 91120 Palaiseau, France
3 Department of Physics, The Ohio State University, Columbus, Ohio 43210, USA
4 Laboratory for Theoretical and Computational Physics, Paul Scherrer Institute,
CH-5232 Villigen, Switzerland
5 Institute of Physics, École Polytechnique Fédérale de Lausanne (EPFL),
CH-1015 Lausanne, Switzerland

* michael.schuler@uibk.ac.at

## Abstract

Anyons in a topologically ordered phase can carry fractional quantum numbers with respect to the symmetry group of the considered system, one example being the fractional charge of the quasiparticles and quasiholes in the fractional quantum Hall effect. When such symmetry-fractionalized anyons condense, the resulting phase must spontaneously break the symmetry and display a local order parameter. In this paper, we study the phase diagram and anyon condensation transitions of a $\mathbb{Z}_2$ topological order perturbed by Ising interactions in the Toric Code. The interplay between the global ("onsite") Ising ($\mathbb{Z}_2$) symmetry and the lattice space group symmetries results in a non-trivial symmetry fractionalization class for the anyons, and is shown to lead to two characteristically different confined, symmetry-broken phases. To understand the anyon condensation transitions, we use the recently introduced critical torus energy spectrum technique to identify a line of emergent 2+1D XY* transitions ending at a fine-tuned (Ising$^2$)* critical point. We provide numerical evidence for the occurrence of two symmetry breaking patterns predicted by the specific symmetry fractionalization class of the condensed anyons in the explored phase diagram. In combination with large-scale quantum Monte Carlo simulations we measure unusually large critical exponents $\eta$ for the scaling of the correlation function at the continuous anyon condensation transitions, and we further identify lines of (weakly) first order transitions in the phase diagram. As an important additional result, we discuss the phase diagram of a resulting 2+1D Ashkin-Teller model, where we demonstrate that torus spectroscopy is capable of identifying emergent XY/O(2) critical behaviour, thereby solving some longstanding open questions in the domain of the 3D Ashkin-Teller models. To establish the generality of our results, we propose a field theoretical description capturing the transition from a $\mathbb{Z}_2$ topological order to either $\mathbb{Z}_2$ symmetry broken phase, which is in excellent agreement with the numerical results.

## 1   Introduction

Quantum phase transitions and quantum critical behaviour are in the center of attention in many areas of science, especially in condensed matter physics. While many quantum phase transitions can be described in terms of the classic Landau-Ginzburg-Wilson theory, in recent years more and more examples of phase transitions beyond this paradigm have been identified [1–4]. A particularly interesting case are phase transitions with an adjacent quantum spin liquid phase, since an increasing number of models featuring such topological phases have been identified in the last years with a multitude of neighboring phases [5–10]. Quantum spin liquids feature fractional excitations with anyonic statistics (anyons) which strongly influence the nature of phase transitions leading to novel universal properties, such as unusu-

ally large critical exponents $\eta$ for the scaling of the correlation function [11–14]. Anyons in a quantum spin liquid can also carry fractional quantum numbers of the symmetry group of the considered system. One prominent example therefore is the fractional charge of the quasiparticles and quasiholes in the fractional quantum Hall effect [15]. When such symmetry-fractionalized quasiparticles condense, it has strong consequences on the fate of the resulting confined phases [16].

The universality class of quantum critical points in quantum spin models is typically identified by precise measurements of critical exponents which define the singular behaviour of observables at criticality. Recently, as an alternative, the analysis of the energy spectrum on a spatial torus at criticality has been explored. It was shown that this critical torus energy spectrum (CTES) is a universal property of critical points and can be used to identify their universality classes [17–21].

The Toric Code [22] is the most prominent quantum spin model which describes a $\mathbb{Z}_2$ topologically ordered (TO) phase. This type of topological order has been realized recently in Rydberg atom arrays [23] and on a superconducting circuit [24]. The Toric Code can be solved exactly, and it therefore forms a patent playground to study topological order. When the Toric Code is perturbed with additional interactions, unconventional phase transitions in quantum spin models can be induced once the topological order is destabilized. The Toric Code in a longitudinal external field [25] with a phase transition between a $\mathbb{Z}_2$ TO phase and a trivial, paramagnetic (*i.e.* non symmetry breaking) phase was successfully studied in Ref. [17] to determine the CTES of the *Ising\** universality class [4, 11], which exhibits unique, characteristic features. The occurrence of such a non-trivial quantum phase transition between these phases naturally leads us to the question what the nature of a direct transition between a $\mathbb{Z}_2$ TO and a topologically trivial, but $\mathbb{Z}_2$ *symmetry-broken* (SB) phase would be [26].

In this paper, we study such a transition in the realm of quantum spin models by perturbing the Toric Code with Ising interactions (TCI). Using a combination of exact diagonalization (ED) and quantum Monte Carlo (QMC) simulations, we observe a rich phase diagram with a $\mathbb{Z}_2$ TO phase and two distinct $\mathbb{Z}_2$ SB phases in the considered parameter regime. The phase transition lines between these phases are found to be of rather varied kind, featuring first-order, a fine-tuned (Ising$^2$)\* [26], and, most prominently, an emergent XY\* transition. In particular, we identify and chart the CTES for the (Ising$^2$)\* and the XY\* transitions and strikingly demonstrate the power of torus spectroscopy to identify emergent symmetries of critical points and the influence of the fractional particles condensing at criticality.

We also identify the non-trivial symmetry fractionalization class of the condensing anyons [16, 27–30] in the TCI model, according to the global $\mathbb{Z}_2$ symmetry group and the lattice space group, which enforces that the condensed phases must spontaneously break the symmetry of the TCI model [16]. In particular, we will show that the condensed phases must either break the spatial symmetry or the spin-inversion symmetry. From numerical simulations, we observe examples for both of these symmetry-breaking patterns in the phase diagram and show that they can be identified from energy level spectroscopy.

From QMC simulations, we also corroborate that the (Ising$^2$)\* and the emergent XY\* transitions feature unusually large critical exponents $\eta^*$. As we will show, the (Ising$^2$)\* transition in our system is particularly appealing, because the large value of $\eta^*$ can be microscopically understood and analytically derived from the known value of $\eta$ of the standard 3D Ising transition [26].

To demonstrate that our study is relevant for generic transitions between $\mathbb{Z}_2$ TO and $\mathbb{Z}_2$ SB phases, we additionally propose a phenomenological field theory description for the transition between such phases. The known fixed points and renormalization group flows of this field theory agree well with the results obtained for our microscopic model. The results presented in this paper are, thus, beyond the specific microscopic model considered here.

As an important additional result obtained along this journey, we discuss a large fraction of the phase diagram of a peculiar resulting (non topological) 2 + 1D quantum Ashkin-Teller model including its quantum phase transitions. We demonstrate that torus spectroscopy allows to characterize the continuous quantum phase transitions with surprising accuracy, given the fact that we only use ED for up to 36 spins. The phase diagram we obtain for the 2 + 1D quantum Ashkin-Teller model resembles the most complex region of the phase diagram of the standard 3D classical Ashkin-Teller model [31–33], where the universality class of critical points along a certain phase transition line is a long debated and unsolved issue [32–36]. Based on universality arguments we believe that the types of phase transitions we obtain in the present work for the 2 + 1D quantum Ashkin-Teller model appear identically in the 3D classical Ashkin-Teller model. Therefore, we plausibly answer this open question in this paper.

In Sec. 2 we introduce the models and numerical methods used in this paper. In particular, we discuss the mapping of the TCI model to an effective Ashkin-Teller transverse field Ising (AT-TFI) model and carefully analyze the symmetries of both models and their relations. Also, we give details on the technique of torus spectroscopy, the QMC method, and advocate a field theoretical description for the considered transition in terms of a two-component $\phi^4$ scalar theory with cubic anisotropy. In Sec. 3 we first thoroughly compute the phase diagram and analyze the phase transitions of the AT-TFI model using torus spectroscopy combined with more standard techniques. In Sec. 4 we finally discuss the phase diagram and phase transitions in the TCI model. To do so, we exploit the microscopic mapping between the TCI and the AT-TFI model and discuss how the presence of fractionalized quasiparticles influences the critical behaviour. We also compute the non-trivial symmetry fractionalization class of the condensing anyons and examine its implications on the possible symmetry breaking patterns in the confined phases. We finally give our conclusions in Sec. 5.

Details of the mapping between the TCI model and the AT-TFI model are shown in App. A. Appendix B provides further information about the QMC algorithm used in this paper. The extrapolation of finite torus energy gaps to obtain the CTES is comprehensively illustrated in App. C, and App. D shows an analysis of the critical exponent $\nu$ in the AT-TFI model from QMC simulations.

# 2 Models & methods

## 2.1 Models

In order to study the transition between a $\mathbb{Z}_2$ TO phase and a $\mathbb{Z}_2$ SB phase we study the Toric Code model as the prime-example of a quantum spin model featuring a $\mathbb{Z}_2$ TO phase and perturb it with Ising interactions. Increasing the strength of the Ising interactions is expected to destroy the $\mathbb{Z}_2$ TO phase at some intermediate coupling strength and to ultimately stabilize a $\mathbb{Z}_2$ spin SB phase for strong Ising couplings. The precise form of the phase diagram and the nature of the phase transitions are some of the prime questions of interest here. For the sake of generality we consider nearest-neighbor as well as next-to-nearest neighbor Ising interactions. The Hamiltonian for this Toric Code-Ising (TCI) model is then given by

$$H = -J_e \sum_s A_s - J_m \sum_p B_p - J_I \sum_{\langle i,j \rangle} \sigma_i^x \sigma_j^x - J_{I_2} \sum_{\langle\langle i,j \rangle\rangle} \sigma_i^x \sigma_j^x , \tag{1}$$

with $A_s = \prod_{i \in s} \sigma_i^x$, $B_p = \prod_{i \in p} \sigma_i^z$, and $J_e, J_m \geq 0$. The Pauli matrices $\sigma_i^\alpha$ describe spins on the links of a square lattice. As shown in Fig. 1(a), $p$ ($s$) denotes plaquettes (stars) on the lattice, $\langle i,j \rangle$ are the nearest neighbors, and $\langle\langle i,j \rangle\rangle$ denote the next-to-nearest neighbor interactions between sites along the edges of the lattice. We will restrict our analysis to ferromagnetic

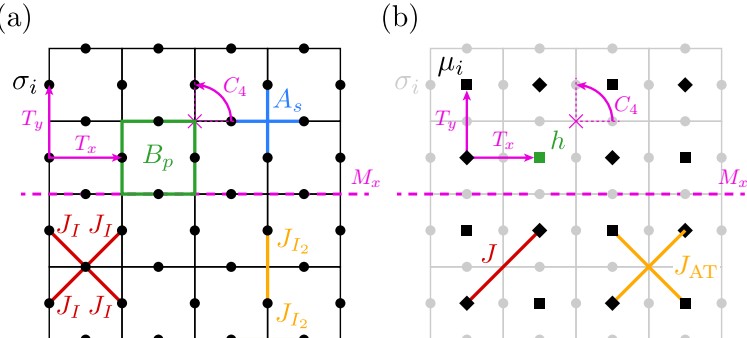

Figure 1: (a) Toric Code model with Ising interactions among nearest neighbor ($J_I$) and next-nearest neighbor sites along the edges of the lattice ($J_{I_2}$), Eq. (1). The spin variable on site $i$ is denoted as $\sigma_i$. The $A_s$ ($B_p$) operators live on stars (plaquettes) of the lattice. In pink, we additionally show the space group symmetry generators: translations $T_{x,y}$, mirror reflection $M_x$, vertex-centered 4-fold rotation $C_4$. (b) Transverse field Ising model with 4-spin interactions, Eq. (2), on the dual square lattice with spin variables $\mu_i$. The original lattice is depicted in light-grey for reference. Squares (diamonds) indicate the sublattice A (B). The mapping of the interactions of the TCI model (a) to the interactions in the AT-TFI model (b) is sketched by using the same colors for mapped interactions. Note that, for better readability, we only illustrate the mapping of one interaction of each type in (b).

nearest neighbor coupling $J_I > 0$, but do not restrict the sign of the next-to-nearest neighbor coupling $J_{I_2}$. We choose Ising interactions along the spin-$x$ direction. The $A_s$ operators then commute with the Ising interactions and are conserved for all values of $J_I$ and $J_{I_2}$.[1] So, we will only consider the charge-free sector $A_s = 1$, $\forall s$, which describes the low-energy physics when $J_e \gg J_m$, such that the other $A_s$ sectors are pushed to high energies. Additionally to the lattice space symmetry group, the Hamiltonian Eq. (1) obeys a global $\mathbb{Z}_2$ spin-inversion symmetry generated by the operator $R_z = \prod_i \sigma_i^z$, as we will elaborate in Sec. 2.2.

In the selected sector ($A_s = 1$, $\forall s$), the TCI model, Eq. (1), can be exactly mapped to a transverse field Ising model on the dual square lattice [17, 25, 37, 38] with additional Ashkin-Teller like four-spin interactions coming from the next-to-nearest neighbor Ising interactions [see Fig. 1(b)], with Hamiltonian

$$H_{\text{AT}} = -h \sum_i \mu_i^z - J \sum_{\langle\langle i,j \rangle\rangle} \mu_i^x \mu_j^x + J_{\text{AT}} \sum_i \mu_i^x \mu_{i+\hat{\mathbf{x}}}^x \mu_{i+\hat{\mathbf{y}}}^x \mu_{i+\hat{\mathbf{x}}+\hat{\mathbf{y}}}^x, \quad (2)$$

$$\text{with} \quad h = J_m, \quad J = 2J_I, \quad J_{\text{AT}} = -2J_{I_2}.$$

The new spins are described by the Pauli matrices $\mu_i^\alpha$ and live on the vertices of the dual square lattice (sites centered on the plaquettes of the original TCI model). $\hat{\mathbf{x}}$ ($\hat{\mathbf{y}}$) denote the unit vectors in $x$ and $y$ direction on the dual lattice. The plaquette operator $B_p$ maps to a transverse field $\mu_i^z$ on each site $i$ of the dual lattice, while the Ising interactions $\sigma_i^x \sigma_j^x$ become either Ising interactions $\mu_i^x \mu_j^x$ or a four-spin interaction involving four $\mu_i^x$ operators at the vertices of a square plaquette. $H_{\text{AT}}$ features an emergent *two-sublattice* structure: The model lives on two interpenetrating square lattices with lattice constant $a' = \sqrt{2}a$, such that the Ising interactions with coupling $J$ only couple spins on the same sublattice [see Fig. 1(b)]. The four-spin interaction with coupling $J_{\text{AT}}$ couples two spins of one sublattice with two spins on the

---

[1]When choosing the Ising interactions along the spin $z$ direction we would simply switch the role of the $A_s$ and $B_p$ operators in the rest of the paper, per the $e-m$ duality of the unperturbed Toric Code.

other sublattice. This emergent *two sublattice* structure is an important difference to the case of perturbing the Toric Code with local magnetic fields studied previously [17, 37, 39], where the resulting TFI model lived on a single square lattice. We call the resulting model, Eq. (2), Ashkin-Teller transverse-field Ising model (AT-TFI).[2]

While we are going to discuss the phase diagram of the AT-TFI model as a stand-alone model without further constraints in Sec. 3, we want to emphasize that the AT-TFI model which one obtains within the $A_s = 1$ sector of the TCI model has a number of constraints and spatial boundary conditions particularities to consider. Specifically, the TCI model, Eq. (1), maps only to the even sector regarding the global Ising symmetry $\mathcal{S} = \prod_i \mu_i^z$ of the AT-TFI model, Eq. (2) [see also Sec. 2.2 for a detailed discussion on the symmetries of both models]. Additionally, the four topological sectors in the $\mathbb{Z}_2$ TO phase of the TCI model, described by eigenvalues $\pm 1$ of Wilson loop operators around the non-contractible paths of the torus, manifest themselves in periodic and anti-periodic boundary conditions for the interactions along the two directions of the torus in the AT-TFI model. Identical boundary conditions have to be chosen for both sublattices, even in the case $J_{AT} = 0$, where the sublattices decouple.

Details of the mapping from the TCI to the AT-TFI model are shown in App. A.

## 2.2  Symmetries

The TCI model, Eq. (1), preserves global ("onsite") symmetries characterized by the group $G_0 = D_2 \times \mathbb{Z}_2^{\mathcal{T}}$, as well as spatial (crystal) symmetries characterized by the wallpaper group $p4gm$. The onsite symmetry group $G_0$ is generated by spin rotations along the $x, y, z$ axes by an angle $\pi$:

$$R_a = \prod_i \sigma_i^a , \quad a = x, y, z \tag{3}$$

and the time reversal symmetry $\mathcal{T} = \prod_i \sigma_i^y \cdot \mathcal{K}$, where $\mathcal{K}$ is the complex conjugation operator. Meanwhile, the space group $p4gm$ is generated by the vertex-centered 4-fold rotation $C_4$, mirror reflection $M_x$ w.r.t. the $yz$ plane parallel to the horizontal nearest-neighbor links, as well as Bravais lattice translations $T_{x,y}$ [see Fig. 1].

As the Ising terms in the TCI model increase and a confinement transition happens, certain symmetries are spontaneously broken. The relevant symmetries across the phase transition are given by the following symmetry group

$$G_s = \mathbb{Z}_2^{R_z} \times p4gm \tag{4}$$

generated by $R_z, C_4, M_x$ and $T_{x,y}$.

Meanwhile, the dual AT-TFI model, Eq. (2), has a larger symmetry group than the original TCI model. In addition to the same crystal symmetry group $p4gm$, it exhibits a $\mathbb{Z}_2 \times \mathbb{Z}_2$ symmetry of flipping all $\mu_i^x$ spins on the two distinct sublattices $A, B$ individually, with symmetry operators

$$\mathcal{S}_{A(B)} = \prod_{i \in A(B)} \mu_i^z . \tag{5}$$

Additionally, each of the spatial symmetry generators can exchange the two sublattices, $A \leftrightarrow B$,

---

[2]We want to note that the structure of the Hamiltonian Eq. (2) is similar to an Ashkin-Teller model [34] where two different spins live on a single site of a square lattice, which are coupled by a 4-body Ashkin-Teller interaction on nearest-neighbor bonds. In our model Eq. (2) the interpenetrating sublattices take the role of the different spins.

yielding the following algebra

$$(\mathcal{S}_A M_x)^2 = \mathcal{S}\,; \tag{6}$$

$$\mathcal{S}_A T_a \mathcal{S}_A^{-1} T_a^{-1} = \mathcal{S}\,, \quad a = x\,, y\,; \tag{7}$$

$$\mathcal{S}_A C_4 \mathcal{S}_A^{-1} C_4^{-1} = \mathcal{S}\,, \tag{8}$$

where

$$\mathcal{S} = \prod_i \mu_i^z = \mathcal{S}_A \mathcal{S}_B \tag{9}$$

is the global Ising symmetry in the AT-TFI model. Mathematically, the symmetry group $G_g$ of the AT-TFI model, generated by $\mathcal{S}_{A,B}$ and $M_x, C_4, T_{x,y}$, is related to the symmetry group $G_s$ of TCI model by the following central extension [27]:

$$1 \to \mathbb{Z}_2 \to G_g \to G_s \to 1\,, \tag{10}$$

where the center $\mathbb{Z}_2$ is generated by the global Ising symmetry $\mathcal{S}$. The algebraic relations Eqs. (6)-(8) suggest that $G_g$ is a nontrivial extension of $G_s$, corresponding to a nontrivial group cohomology $\mathcal{H}^2(G_s, \mathbb{Z}_2) \neq 0$. As will be discussed later in Sec. 4.2, this is a direct consequence of the nontrivial symmetry fractionalization class [27–30] for $m$ particles in the TCI model, and has strong implications on the spontaneous symmetry breaking patterns across the phase boundary [16].

Microscopically, the TCI model, Eq. (1), maps only to the even sector regarding the global Ising symmetry $\mathcal{S}$ of the AT-TFI model Eq. (2). Physically, $\mathcal{S}$ counts the parity of the total number of $m$ particles in the TCI model, which is always even on a torus. On the other hand, the global $\mathbb{Z}_2$ symmetry $R_z$ of the TCI model maps to the sublattice inversion symmetries of the AT-TFI model, $R_z = \mathcal{S}_A = \mathcal{S}_B$. It is important to note here, that $\mathcal{S}_A$ and $\mathcal{S}_B$ are independent symmetries in the AT-TFI model; only through the constraint of global evenness are the symmetries $\mathcal{S}_A$ and $\mathcal{S}_B$ bound to be equivalent in the TCI model. Mathematically, in the exact sequence Eq. (10), $R_z$ in $G_s$ has two preimages in $G_g$, $\mathcal{S}_A$ and $\mathcal{S}_B$, in the surjective map $G_g \to G_s$.

## 2.3 Phenomenological quantum field theory description

We can find an effective quantum field theory describing the properties of the AT-TFI model in the vicinity of the sublattice decoupled point $J_{AT} = 0$. To do so, we consider two scalar fields $\phi_1$, $\phi_2$, one for each of the sublattices, which are described by the standard Landau-Ginzburg-Wilson (LGW) theory. The LGW Hamiltonian is chosen such that it is invariant under the $\mathbb{Z}_2 \times \mathbb{Z}_2$ symmetry of flipping the sign of both fields individually ($\phi_1 \to -\phi_1$, $\phi_2 \to -\phi_2$), and under the exchange of the two fields ($\phi_1 \leftrightarrow \phi_2$) because of the equivalence of sublattices in the AT-TFI model (defined by the symmetry operator of exchanging the sublattices $\mathcal{S}_{A \leftrightarrow B}$). The interaction among the two different fields (coupling of sublattices in the AT-TFI model for $J_{AT} \neq 0$) can be modelled by a term $\phi_1^2 \phi_2^2$ which is the lowest order interaction term respecting the symmetries. The resulting field theory is the $n = 2$-component LGW Hamiltonian with cubic anisotropy in $D = (2 + 1)$ dimensions [40–42], where we only include scaling relevant terms

$$\mathcal{H} = \int d^2 x \sum_{i=1}^{2} \frac{1}{2} \left( \Pi_i^2 + (\nabla \phi_i)^2 + r \phi_i^2 + \frac{u}{12} \phi_i^4 \right) + \frac{v}{4!} \phi_1^2 \phi_2^2\,. \tag{11}$$

A renormalization group analysis of Eq. (11), with a $n = 2$ component scalar field in $D = (2 + 1)$ space-time dimensions features, apart from unstable Gaussian and cubic fixed points, a stable O($n$) symmetric fixed point, an unstable Ising fixed point (of $n$ identical Ising

fields), and, outside the attraction regime of the fixed points, a flow towards first order behaviour [41, 42]. The Ising fixed point can only be reached by fine-tuning of two parameters. We will show numerically in this paper, that the AT-TFI model indeed features an extended critical line in the 3D O(2) universality class, a line of (weakly) first order transitions and a so-called *Ising²* critical point, which can only be reached by fine-tuning both parameters $J_{\mathrm{AT}}/J$ and $h/J$. The cubic anisotropy model, Eq. (11), is thus a good description of the AT-TFI model around $J_{\mathrm{AT}}/J = 0$ in the infrared scaling limit.

The cubic anisotropy model also describes the transition between a $\mathbb{Z}_2$ TO and a $\mathbb{Z}_2$ SB state, which we eventually want to discuss in this paper in the context of the TCI model, when certain properties for the fields are enforced. This is analogous to the transition between a $\mathbb{Z}_2$ TO phase and a paramagnetic phase, which is described by the Ising* field theory, *i.e.* the Ising field theory with additional constraints for the scalar field [11, 13, 17]. The transition from the $\mathbb{Z}_2$ TO phase is induced here by the condensation of the (fractional) $m$ particles of the spin liquid. We describe the $m$ particles by a scalar field $\phi$ which must be composed of two real components to carry a $\mathbb{Z}_2$ charge, *i.e.* $\phi = (\phi_1, \phi_2)$, and the model must be invariant under $\phi_1 \leftrightarrow \phi_2$. Also, as $m$ particles can only be created in pairs, the Hamiltonian may only contain terms even in $\phi$. The Hamiltonian, where we again consider only scaling relevant terms, is then precisely the cubic anisotropy model, Eq. (11). The fractional nature of the $m$ particles has further consequences: Since $m$ particles can only be created in pairs, all physical observables are at least bilinear combinations of the fields $\phi_i$. Additionally, the boundary conditions of the torus have to be generalized to be both, periodic (P) and anti-periodic (A), along both directions of the torus, as the fields $\phi$ and $-\phi$ are physically indistinguishable. The distinct boundary conditions represent the different topological sectors in the TO phase. These additional constraints for the fields correspond to the microscopic constraints of even global spin-inversion symmetry and (anti-)periodic boundary conditions observed in the microscopic mapping from the TCI to the AT-TFI model.

## 2.4 Numerical methods

We use a combination of exact diagonalization (ED) and quantum Monte Carlo (QMC) [43] simulations to explore the phase diagram and, in particular, the quantum critical features of the TCI model, Eq. (1). For this purpose, we first chart the phase diagram of the topologically trivial AT-TFI model, Eq. (2), in the ground state sector of periodic boundary conditions, and then use the microscopic mapping between the TCI and AT-TFI model to investigate properties of the TCI model in a later section.

We use discrete-time QMC simulations of the AT-TFI model which maps the partition function into an anisotropic 3D Ising model with Ising and Ashkin-Teller couplings within an imaginary-time slice, and Ising couplings between the time slices [see App. B for more details on the QMC method]. The different phases appearing in the AT-TFI model can be uniquely distinguished by measuring the equal-time sublattice magnetization histogram $P(m_A, m_B)$, which is defined as the probability to find sublattice $A$ ($B$) with total magnetization $m_A = \langle \sum_{i \in A} \mu_i^x \rangle$ ($m_B = \langle \sum_{i \in B} \mu_i^x \rangle$) within a given imaginary-time slice of the sampled ground state configurations. From $P(m_A, m_B)$ any two-sublattice order parameter can be extracted, such as $\langle m_S \rangle = \langle |m_A| + |m_B| \rangle$, which is non-zero in the ordered phases we will obtain below, but becomes zero in the paramagnetic phase (PM). The precise location of the phase boundary between ordered and disordered phases can be obtained from the Binder ratio $U = \langle m_S^4 \rangle / \langle m_S^2 \rangle^2$ [44]. $U$ is independent of the system size (up to higher-order corrections) at critical points, such that the location of a critical point can be identified from a crossing of $U$ for different system sizes. We perform QMC simulations on periodic square clusters of size $N = L \times L$ with $L \leq 48$ to accurately estimate the location of phase transition points and their properties.

Around critical points, we also compute the four-point correlation function

$$C(r) = \langle \mu_{0,A}^x \mu_{0,B}^x \mu_{r,A}^x \mu_{r,B}^x \rangle, \tag{12}$$

which is related to the two-point function of the $\sigma$ spin operators of the TCI model, and its corresponding correlation length $\xi$. Here $\mu_{i,A/B}^x$ denotes the Pauli-$x$ operator on one of the two plaquettes (dual lattice) neighboring to lattice site $i$ (original lattice) and $A/B$ labels the corresponding sublattice of the dual lattice [see App. B for more details]. A standard finite-size scaling analysis [44] can then be used to estimate the values of the critical exponents $\eta$ and $\nu$, which are universal properties of the quantum critical point (see below for details). Here, we should mention, that the QMC simulations of the AT-TFI model include all spin-inversion sectors and do not explicitly consider the additional constraints (only even spin-inversion sectors, different boundary conditions) in the mapping to the TCI model. Since we are interested in ground state physics at very low temperatures, the additional presence of the spin-inversion odd levels and the absence of the other boundary condition sectors (corresponding to the different topological sectors) should not influence the results significantly for the TCI model.

Our main approach to identify the nature of the quantum critical points in the TCI and AT-TFI models is the recently introduced technique of measuring the CTES [17–21]. CTESs are universal properties of quantum critical points. Thus, they can be used to identify the universality class of a critical point under investigation by simply comparing its CTES to already charted ones. In contrast to critical exponents, CTESs have been found to be qualitatively very different among different universality classes. In particular, the sequence and number of the low-lying CTES levels and their multiplicities vary strongly, such that finite-size clusters of a few ten spins are typically already sufficient to uniquely identify them.

The CTES for a critical point is computed with ED: We tune the Hamiltonian parameters to the critical values, estimated by extrapolating the crossings of the Binder cumulant to the thermodynamic limit. We then compute the low-energy spectrum on finite-size systems of $N \le 36$ spins on a spatial torus. The $i$-th energy gap to the ground state is denoted as $\Delta_i^N$. Due to the translational symmetry the energy levels can be labelled by a quantum number $\kappa = \sqrt{N}/(2\pi)|\mathbf{k} \bmod \mathrm{M}|$ corresponding to the momentum $\mathbf{k}$ of the energy eigenstate, and $\mathrm{M} = (\pi, \pi)$.[3] We will only consider the most important $\kappa = 0$ spectrum in this paper. Additionally, each level carries a quantum number even/odd ($s_i = \pm 1$) according to the global spin-inversion symmetry $\mathcal{S}$. We do not measure the quantum numbers of the sublattice inversion symmetries $\mathcal{S}_{A,B}$. The quantum number according to the exchange symmetry of the two sublattices $\mathcal{S}_{A \leftrightarrow B}$ is encoded in the irreducible representation of the square lattice space group. In particular, a state with momentum $\mathbf{k} = \Gamma = (0, 0)$ is even, while a state with momentum $\mathbf{k} = \mathrm{M}$ is odd under the sublattice exchange symmetry $\mathcal{S}_{A \leftrightarrow B}$, for the trivial irreducible representation of the lattice point group ($C_4$ or $D_4$).

We multiply the bare low-energy gaps with the linear system size $\sqrt{N}$ to get rid of the dominant scaling factor and extrapolate each level $\Delta_i^N \times \sqrt{N}$ to the thermodynamic limit $N \to \infty$ to obtain the CTES levels, which we denote as $\Delta_i \times \sqrt{N}$ [4] [see also Fig. 4(b) and Appendix C]. We only track the lowest energy gaps. They form the most important levels of the CTES and can be phenomenologically related to the relevant fields of the underlying critical field

---

[3]In the context of the AT-TFI model the ordered phases spontaneously break symmetry by building a superposition of states with momenta $\mathbf{k} = \Gamma = (0, 0)$ and $\mathbf{k} = \mathrm{M} = (\pi, \pi)$. The dispersion relation of the quasi-particle thus has minima around $\Gamma$ and M which become effective light cones at the critical point. $\kappa = 0$ thus contains levels with momenta $\Gamma$ and M.

[4]We omit giving error bars on the CTES values throughout the paper. Since the precise form of the subleading finite-size corrections to the CTES levels is not known, it is difficult to estimate a reasonable error from the extrapolation scheme. Also, the qualitative structure of the CTES is its most valuable property to identify universality classes, and not their precise values.

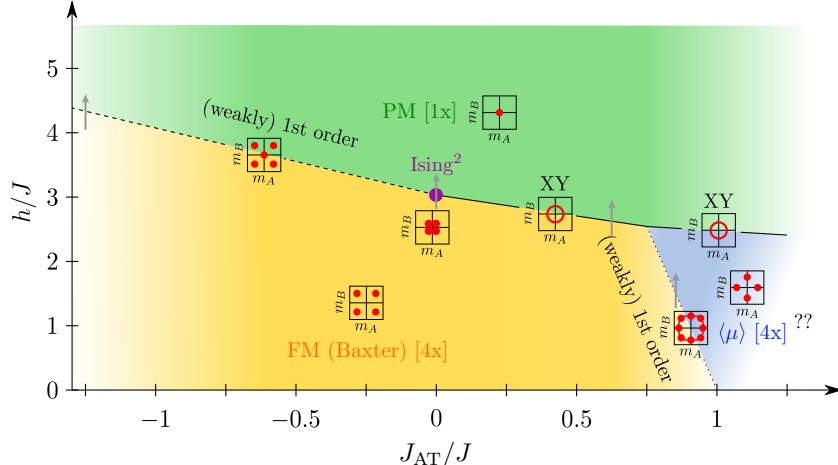

Figure 2: Sketch of the phase diagram for the AT-TFI model, Eq. (2). Full lines denote continuous phase transitions, dashed lines (weakly) first order transitions. The histograms sketch the typical sublattice magnetization histograms $P(m_A, m_B)$ for the corresponding phases and at the phase transitions. The magenta point denotes the Ising$^2$ critical point, XY denotes criticality in the 3D XY/O(2) universality class. The question marks indicate regions we have not investigated in detail. The gray arrows indicate the paths through the phase transitions shown in Fig. 3. The ground state degeneracy of the phases is given by the numbers in square brackets.

theory [17, 19]. They would also show up as the lowest energy levels in the spectrum on a sphere [45].

CTESs are given by universal numbers $\xi_i$ times a non-universal constant $c$ corresponding to the effective speed of light which depends on microscopic parameters, *i.e.* $\Delta_i \times \sqrt{N} = c\,\xi_i$. In this paper, we do not compute the effective speed of light $c$. Therefore, we typically normalize the CTESs by the first non-zero level to compare them among each other. To identify the universality classes of critical points in the AT-TFI model, we compare the measured CTESs along phase boundaries to already charted Wilson-Fisher CTESs [19]. This approach provides a powerful complementary method to the standard identification scheme for universality classes based on measuring critical exponents, as we will demonstrate in this paper.

## 3 Analysis of the AT-TFI model

We begin our discussion with a thorough analysis of the unconstrained AT-TFI model using periodic boundary conditions. While this analysis is interesting in itself, we will eventually also be able to infer many properties of the original TCI model in Sec. 4, using the microscopic mapping and its constraints and boundary conditions discussed above.

### 3.1 Phase diagram

The different phases appearing in the AT-TFI model can be uniquely distinguished by the sublattice magnetization histogram $P(m_A, m_B)$ [see Sec. 2.4]. We compute $P(m_A, m_B)$ with ED on clusters of a few ten spins and with QMC for clusters up to 32 x 32 spins. The so obtained phase diagram of the AT-TFI model is sketched in Fig. 2 and discussed in the following.

In the special case $J_{AT} = 0$, the properties of the AT-TFI model can be easily understood. The two sublattices $A$, $B$ of the square lattice decouple and Eq. (2) decomposes into two identical

copies of a ferromagnetic transverse field Ising (TFI) model, each on a single sublattice. For small transverse fields $h/J$ the TFI models exhibit ferromagnetic order, individually on both sublattices. The sublattice magnetization histogram therefore shows four peaks at the locations $m_{A,B} = \{(+m, +m), (-m, +m), (+m, -m), (-m, -m)\}$, where $m > 0$ denotes the magnetization of the standard TFI model for this value of $h/J$ [see Fig. 3(a)]. This state is often referred to as the Baxter ferromagnet (FM) state in the literature on Ashkin-Teller models [34, 35, 46]. Note, that this FM state contains not only states where all spins are parallel, but also ones, where the spins on the different sublattices are antiparallel, in contrast to the standard Ising FM state. For large $h/J$ the two TFI models simultaneously become disordered (paramagnetic, PM) and $P(m_A, m_B)$ shows a Gaussian peak centered around zero magnetization $m_{A,B} = 0$. The phase transition happens precisely at the critical point of the TFI model $(h/J)_c^0 \approx 3.04438(2)$ [43].

We will now continue discussing the generic case $J_{AT} \neq 0$, which couples the two sublattices through a four-spin term. We find that both the FM and PM phases are stable against such a perturbation and form extended phases. The PM phase extends throughout the entire considered $J_{AT}$ range. The exact position of the phase transition line $(h/J)_c$ to the other phases, however, is modified by $J_{AT}$.

For ferromagnetic four-spin interactions, $J_{AT} < 0$, the FM phase is stable up to at least $J_{AT}/J \approx -1.3$. We have not investigated the phase diagram further in this direction and leave this for future studies. For antiferromagnetic $J_{AT} > 0$, the FM phase is destabilized at $J_{AT}/J \approx 1$, where a transition to another ordered phase, which we denote $\langle \mu \rangle$, is found. In the $\langle \mu \rangle$-phase only one of the two sublattices is ferromagnetically ordered, while the magnetization of the other sublattice is zero. The histogram shows peaks at the distinct locations $m_{A,B} = \{(+m, 0), (-m, 0), (0, +m), (0, -m)\}$ [see also Fig. 3(d), right plot].

We suppose that the $\langle \mu \rangle$ phase derives from an order-by-disorder mechanism [47–49] induced by the quantum fluctuations of the transverse field: At $h = 0$ we observe a huge number of (classical) ground states with identical energy for $J_{AT}/J \geq 1$. The number of these classical states increases with system size $N$. Any finite transverse field $h/J > 0$ gaps out most of these states, except for four states which remain very low in energy and correspond to the $\langle \mu \rangle$ phase [see also Sec. 3.2.3]. While we have not investigated in detail if the zero temperature entropy density at $h = 0$ is indeed macroscopically large, our results indicate that the $\langle \mu \rangle$ phase comes from an order-by-disorder mechanism.

The $\langle \mu \rangle$ phase becomes unstable rather quickly when $J_{AT}$ is increased further and the AT-TFI model becomes strongly frustrated. We then observe a large number of low-lying energy levels in the energy spectrum in multiple different symmetry sectors even at finite $h/J$ [see also Fig. 6 below]. We leave an investigation of this parameter regime for future studies.

## 3.2 Phase transitions

In the following, we will discuss in detail the different types of quantum phase transitions that appear in the AT-TFI model. The precise locations of the phase boundaries are computed through binder ratios of the sublattice order parameter using QMC [see Sec. 2.4]. In Fig. 3 we first show sublattice magnetization histograms from QMC for different cuts through phase boundaries in the AT-TFI model in the considered parameter range. This data already allows for a first qualitative analysis of the different types of phase transitions appearing in this model.

In the sublattice decoupled case, $J_{AT} = 0$, the four peaks along the diagonals (FM) continuously transform into a broader central peak (PM) when the transverse field $h/J$ is increased [see Fig. 3(a)]. This confirms our expectation that both sublattices individually, but simultaneously undergo a standard 2+1D Ising ($\mathbb{Z}_2$) quantum phase transition at the critical point $(h/J)_c^0$. We label the critical point at $J_{AT} = 0$ as $Ising^2$, for reasons becoming clear later.

For negative $J_{AT}/J < 0$ [see Fig. 3(b)], the FM peaks in the histogram do not transform smoothly to a central peak. Rather, we observe a clear coexistence of the four peaks along the

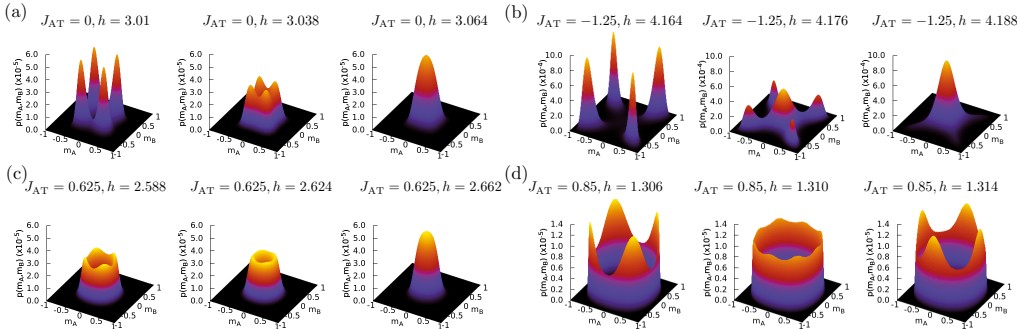

Figure 3: Sublattice magnetization histograms $P(m_A, m_B)$ for paths through different types of phase transitions. For each path, we fix $J_{AT}$ and vary $h$ to cross the transition. The left and right panels show the histograms within the corresponding phases, the central panels are very close to the phase transition point. For all plots $L = 32$ except for (b) where $L = 16$ is shown. (a) Ising$^2$ transition from the FM phase to the PM phase for $J_{AT} = 0$. The central panel shows that the four peaks (FM) smoothly transform into a single peak (PM), illustrating the continuous Ising$^2$ nature of the critical point. (b) First order transition between the FM and the PM phase for $J_{AT}/J = -1.25$. The central panel shows phase coexistence of the two phases where the four peaks at the diagonal (FM) are equally present with the center peak (PM). (c) XY/O(2) transition from the FM to the PM phase for $J_{AT}/J = 0.625$. Around the critical point (central panel) the histogram becomes rotationally symmetric illustrating the emergent O(2) symmetry of the critical point. (d) (Weakly) first order transition between the FM and the $\langle \mu \rangle$ phase for $J_{AT}/J = 0.85$. The central panel shows that the absolute magnetization $\langle m_S \rangle$ is finite at the transition. The distinct four plus four peaks characteristic for both phases are weak, but equally present.

diagonal with another peak in the center. This sign of phase coexistence of the FM and PM at the phase transition is an indicator for a first-order phase transition. For the available system sizes phase coexistence can only be uniquely identified far enough away from the termination point of the first order transition line at $J_{AT}/J = 0$. The first order transition can be made arbitrarily weak when $J_{AT}/J$ approaches $J_{AT}/J = 0$ from below.

For positive $J_{AT}/J > 0$ [see Fig. 3(c)], the histogram around the phase transition looks again different. It smoothly transforms into a circle and becomes emergently rotational symmetric when the critical point is approached from within the ordered phase, while its radius is continuously decreasing. This rotational O(2) symmetry is an emergent feature at criticality since the Hamiltonian, Eq. (2), is not invariant under this continuous symmetry group, and hints at an emergent XY/O(2) criticality [2, 50] for the phase transition between the FM and the PM when $J_{AT}/J > 0$. We corroborate this finding using the CTES below.

Finally, let us consider the transition between the two ordered phases, FM and $\langle \mu \rangle$ [see Fig. 3(d)]. The sublattice magnetization histogram then retains a finite radius at the phase transition and a coexistence of the four (FM) plus four ($\langle \mu \rangle$) peaks is observable. This coexistence together with the non-vanishing (with system size) values of the order parameters (proportional to the radius of the histogram) strongly indicates a (weakly) first-order transition between the ordered phases.

After this informative first characterization of the phase transitions we will, in the following, analyze those in more detail. To characterize the critical properties of the continuous quantum phase transitions, we focus on the novel approach of measuring the CTES with ED. We complement our results with the more traditional approach of estimating critical exponents from QMC simulations.

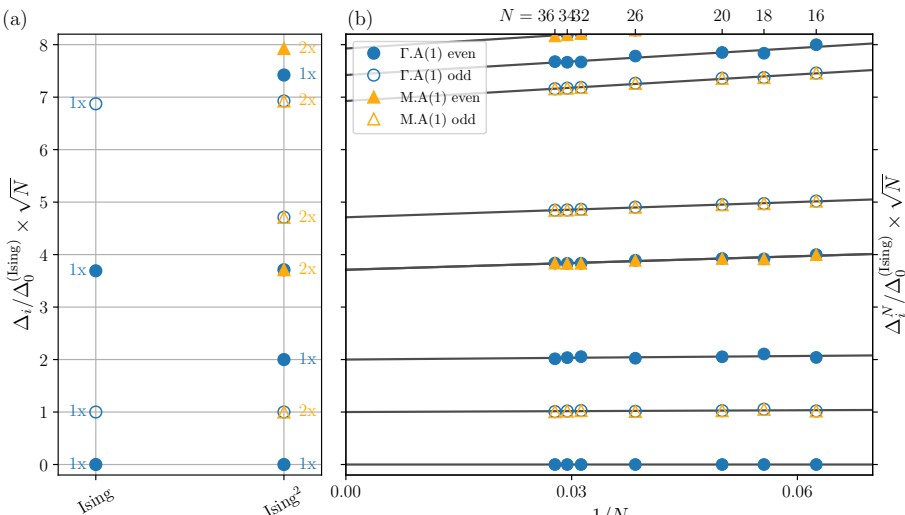

Figure 4: CTES for the Ising$^2$ transition at $J_{AT} = 0$ for $\kappa = 0$ levels only. The CTES is normalized such that the first gap is set to unity. Circles (triangles) denote the momentum sectors $\mathbf{k} = \Gamma = (0,0)$ ($\mathbf{k} = \mathrm{M} = (\pi, \pi)$), while filled (open) symbols indicate levels even (odd) under global spin inversion symmetry $\mathcal{S}$. (a) Ising$^2$ CTES compared to the already charted Ising CTES [17]. Each combination of two levels of the Ising CTES gives a level in the Ising$^2$ CTES, where the gap is the sum of the gaps in the Ising CTES. The numbers denote the full multiplicities of the levels. (b) Finite-size extrapolations of the scaled energy gaps $\Delta_i^N \times \sqrt{N}$ of the AT-TFI model at $J_{AT} = 0$. We perform linear expansions of scaled finite-size energy gaps (symbols) in $1/N$ (black lines) to obtain the Ising$^2$ CTES for $N \to \infty$.

### 3.2.1 Ising$^2$

First, let us again consider the illustrative case $J_{AT} = 0$, where the two sublattices of the AT-TFI model decouple. As we have discussed, tuning the parameter $h/J$ drives the AT-TFI model through a quantum critical point, where both sublattices, individually but simultaneously, undergo a transition in the Ising universality class, *i.e.* the full system undergoes a Ising$^2$ transition. We choose the label Ising$^2$ because the CTES is sensitive to the presence of *two* copies of the Ising critical point, as we will discuss below.

We measure the CTES for the Ising$^2$ transition by computing the low-energy spectrum for finite systems with ED. We extrapolate the individual energy levels $\Delta_i^N \times \sqrt{N}$ linearly in $1/N$ to the thermodynamic limit $N \to \infty$ to obtain the CTES, as shown in Fig. 4(b) [see Sec. 2.4 for further details]. The (normalized) Ising$^2$ CTES shown in Fig. 4(a) is then given by these extrapolated values $\Delta_i \times \sqrt{N}$ together with the quantum numbers and degeneracies of the corresponding eigenstates, and is a universal property of the universality class corresponding to the critical point. Due to the microscopic sublattice decoupling, the Ising$^2$ CTES can also be directly obtained from suitable combinations of the CTES of the standard 2+1D Ising transition [17]. In particular, each combination of two levels in the Ising CTES yields a level in the Ising$^2$ CTES with the gap equal to the sum of the single Ising gaps. The quantum numbers and degeneracies can also be inferred directly. We show a comparison between the Ising and Ising$^2$ CTES in Fig. 4(a). Note that the Ising$^2$ CTES contains two low-lying $\mathbb{Z}_2$ even, momentum $\Gamma$, levels above the vacuum (at rescaled energies 2 and $\sim 3.8$), hinting at the fact that the Ising$^2$ critical point is multicritical, i.e. requires fine-tuning of two parameters to be reached. This is in strong contrast to the standard Ising CTES which is not multicritical and only contains a single low-lying $\Gamma$ even level (at rescaled energy $\sim 3.8$).

Complementary, we analyze the correlation length $\xi$ across the Ising$^2$ transition in a standard finite-size scaling approach using QMC data. We find that the data is consistent with the 3D Ising critical exponent $\nu_{\text{Ising}} = 0.629971(4)$ [51], as expected for the Ising$^2$ universality class, where the two sublattices decouple [see App. D].

The CTES and standard finite-size scaling analysis thus agree on the identification of the Ising$^2$ universality class for the phase transition at $J_{\text{AT}} = 0$.

### 3.2.2 Generic FM – PM transition

In this section, we will show that the coupling between the two sublattices for the generic case $J_{\text{AT}} \neq 0$ will drastically alter the critical behavior. The multicritical Ising$^2$ transition can only be reached in the fine-tuned case of decoupled sublattices, $J_{\text{AT}} = 0$.

As described above, for $J_{\text{AT}}/J < 0$ we find coexistence of the FM and the PM phase at the phase transition line in the sublattice magnetization histograms [see Fig. 3(b)]. Additionally, the Binder cumulant shows a sharp negative peak around the phase transition point [not shown], which is another important sign of a first order transition. Due to the limited system sizes, we can clearly observe the phase coexistence only for negative $J_{\text{AT}}$ far enough from $J_{\text{AT}} = 0$. Closer to this fine-tuned case a slow crossover flow from the Ising$^2$ critical point to the (weakly) first-order transition appears. The crossover length scale becomes increasingly large when $J_{\text{AT}}$ is tuned closer to the Ising$^2$ transition and larger system sizes would be necessary to directly observe phase coexistence, i.e. the first-order transition can be made arbitrarily weak. From field theoretical arguments, in particular the RG flow analysis of the cubic anisotropy model [41, 42] shortly summarized in Sec. 2.3, we expect that the entire phase transition line between the FM and the PM is first order for ferromagnetic $J_{\text{AT}} < 0$.

For $J_{\text{AT}}/J > 0$ yet another type of transition between the Baxter FM and PM phases is found. Binder cumulant analysis and the investigation of the sublattice magnetization histograms indicate a continuously vanishing (sublattice) magnetization with emergent rotational O(2) symmetry [see Fig. 3(c)]. Thus, the phase transition is expected to be continuous, possibly giving rise to a line of quantum critical points in the 3D XY/O(2) universality class. The transition from the $\langle \mu \rangle$ to the PM phase, for even larger $J_{\text{AT}}/J$, shows similar emergent O(2) symmetry of the histograms. We, therefore, also predict XY criticality for the transition between these phases.

Now, we want to highlight, that the emergence of XY critical behaviour can be uniquely identified from a careful analysis of the CTES with systems of only a few ten sites, which is one of the main results of this paper. In Fig. 5 we show the (normalized) CTES along the critical line for $J_{\text{AT}}/J > 0$ [see also App. C], and for the established Ising$^2$ case, $J_{\text{AT}}/J = 0$, as a comparison. For $J_{\text{AT}}/J = 0.625$ (0.95) the transition is between the PM and the Baxter FM ($\langle \mu \rangle$) phase, respectively. The parameter $J_{\text{AT}}/J = 0.75$ is very close to the point where the XY critical line and the (weakly) first order line (between the FM and $\langle \mu \rangle$ phases) meet [see also Fig. 2]. For $J_{\text{AT}}/J > 0$ the spectrum changes considerably compared to the Ising$^2$ CTES. In particular, singly degenerate levels recombine and build new sets of two-fold (quasi-)degenerate or single levels as indicated by the arrows in Fig. 5. On the right hand side of Fig. 5 we additionally plot the previously charted [19], normalized CTES for the 2+1D XY/O(2) critical point in the $\kappa = 0$ sector. These levels can be labeled by a quantum number $S_z$ and are non-degenerate when $S_z = 0$ and doubly degenerate for $S_z > 0$. Comparing the to be classified CTESs for $J_{\text{AT}} > 0$ with this XY CTES, we find a compelling agreement: not only do the quantitative values of the gaps agree well, but, in particular, the (quasi-)degeneracies of the levels are the ones predicted from the XY CTES [see e.g. the $S_z = 2$ level and the second $S_z = 0$ level]. Also, the even/odd quantum numbers of the levels, regarding the global $\mathbb{Z}_2$ spin-inversion symmetry of the AT-TFI model, agree with the even/odd $S_z$ sectors of the XY/O(2) levels. Therefore, the CTES for the FM – PM and the $\langle \mu \rangle$ – PM transitions at $J_{\text{AT}}/J > 0$ emergently matches the XY CTES, which

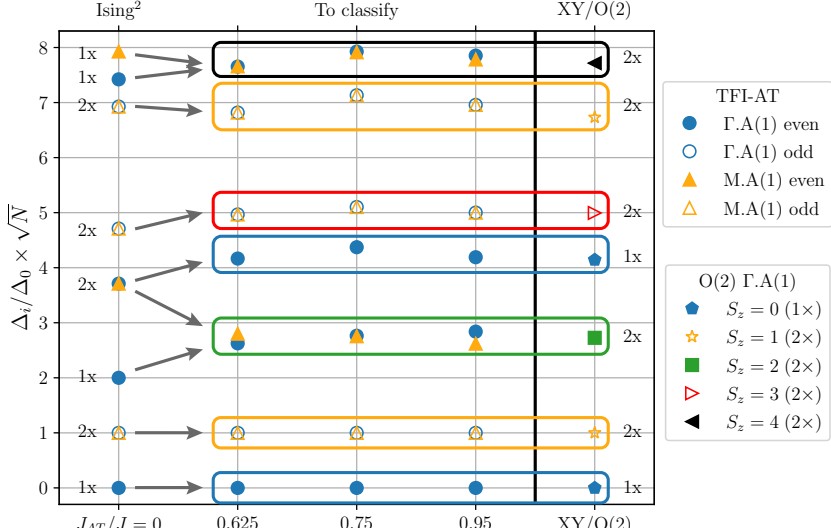

Figure 5: Evolution of the CTES from the Ising$^2$ ($J_{AT} = 0$) to emergent XY/O(2) criticality ($J_{AT}/J > 0$). As a comparison, we show the CTES for the standard XY/O(2) theory at $\kappa = 0$ [see Ref. [19]]. The levels of the Ising$^2$ transition recombine to emergently match the energies and degeneracies of the standard XY/O(2) critical spectrum as indicated by the arrows and boxes, while becoming strongly different from the Ising$^2$ critical spectrum. Even (odd) denote $s_i = 1 (-1)$ levels under global spin-inversion symmetry $\mathcal{S}$ and are shown with full (empty) symbols for the AT-TFI model. The numbers on the left (right) denote the quasi-multiplicities of the CTES levels in the Ising$^2$ (XY) CTES, respectively.

is a highly non-trivial feature, and strongly suggests that the continuous phase transition line has an emergent O(2) symmetry and belongs to the 3D XY/O(2) universality class.

The CTES levels for $J_{AT}/J > 0$ which correspond to the $S_z = 2$ XY/O(2) field [see green box in Fig. 5] show a small splitting in energy: for $J_{AT}/J = 0.625$ the $\Gamma$ even level (blue circle) is the lower energy level, while for $J_{AT}/J = 0.95$ the M even level (yellow triangle) is the lower one. The lower of these levels, together with the three further lower energy levels, *i.e.* the ones corresponding to $S_z = 0$ and $S_z = 1$, form the four-fold quasi-degenerate ground state manifold for the adjacent ordered FM and $\langle \mu \rangle$ phases in the thermodynamic limit, respectively. The small splitting of the $S_z = 2$ levels can now be interpreted as resulting from dangerously irrelevant terms in the critical 3D XY theory. They do not influence critical properties but eventually drive the PM into distinct symmetry-broken phases. In other words, we can read off the sign of the (dangerously irrelevant) $q = 4$ monopole operator coupling constant at the 3D XY fixed point from the order of the two $S_z = 2$ XY levels in the CTES. The sign change for this operator along the XY transition line in the AT-TFI model happens approximately at $J_{AT}/J \approx 0.75$, where the splitting of the $S_z = 2$ levels vanishes. Incidentally, this is the location where the (vertical) FM – $\langle \mu \rangle$ transition line meets the (horizontal) 3D XY line in Fig. 2. We will come back to this observation below in Sec. 3.2.3. Finally, we should mention that the finite-size energy spectra along the phase boundary line evolve smoothly from the Ising$^2$ case to the XY case for a given system size. With the limited system sizes available, we need to choose a sufficiently large value of $J_{AT}/J$ to overcome the crossover length scale and to observe the clean characteristic XY CTES. So, in the immediate vicinity of the Ising$^2$ transition we are not in a position to detect the 3D XY behaviour using torus spectroscopy on clusters of only a few ten sites, even though this critical behaviour is expected.

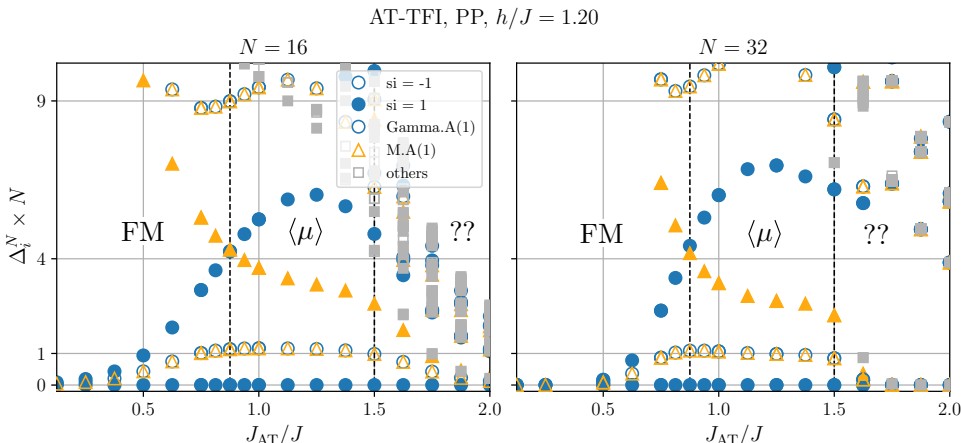

Figure 6: Low-energy spectrum for the AT-TFI model as a function of $J_{AT}/J$ for a fixed $h/J = 1.2$ on clusters of $N = 16$ (left panel) and $N = 32$ (right panel) spins. The energy levels are characterized by their momentum $\Gamma = (0,0)$ or $M = (\pi, \pi)$ and their quantum number even ($s_i = 1$, full symbols) or odd ($s_i = -1$, empty symbols) under global spin inversion symmetry. The four lowest energy levels for small $J_{AT}/J \lesssim 0.875$ correspond to the prediction for the FM phase, while for $J_{AT}/J \approx 0.875 - 1.5$ the four lowest levels indicate the $\langle \mu \rangle$ phase (see text). For $J_{AT}/J \gtrsim 1.5$ the AT-TFI model becomes strongly frustrated and the $\langle \mu \rangle$ phase is destabilized. At the weakly first order transition $J_{AT}/J \lesssim 0.875$ the finite-size spectra approximate an O(2) rotor spectrum for these system sizes (see text).

Additionally, we perform a more standard finite-size analysis of the correlation length $\xi$ across the phase transition, exemplarily for $J_{AT}/J = 0.625$, which is consistent with the known critical exponent for the 3D XY universality class, $\nu_{XY} = 0.6719(11)$ [51] [see App. D].

The different types of phase transitions between the PM and the FM obtained here around $J_{AT}/J = 0$ are also supported by a renormalization group analysis [41, 42] of the related cubic anisotropy LGW model, Eq. (11) [see Sec. 2.3]. For a $n = 2$ component scalar field in $D = (2 + 1)$ space-time dimensions it features, apart from unstable Gaussian and cubic fixed points, a stable O($n$) symmetric fixed point, an unstable Ising fixed point (of $n$ identical Ising fields), and, outside the attraction regime of the fixed points, a flow towards first order behaviour [41, 42]. The stable O($n$) fixed point corresponds to the extended O(2) critical line observed for $J_{AT}/J > 0$ in the AT-TFI model. The flow towards first order transitions for parameters outside the attraction regime of the fixed points corresponds to the line of first order transitions between the PM and FM phases for $J_{AT}/J < 0$. The unstable Ising fixed point corresponds to the Ising$^2$ transition, which can, accordingly, only be reached by fine-tuning the model parameters, i.e. adjusting $J_{AT}/J = 0$ and $h/J = (h/J)_c^0$, simultaneously.

To sum up, we find good agreement of the critical properties between the FM and the PM phases among all three complementary approaches: CTES, critical exponent analysis, and LGW field theory.

### 3.2.3 FM – $\langle \mu \rangle$ transition

The transition between the two ordered phases, FM and $\langle \mu \rangle$, is found to be (weakly) first order from an analysis of the sublattice magnetization histograms, as discussed above [see also Fig. 3(d)].

Since the transition between these ordered phase is not continuous it does not feature a well-defined CTES. Yet, both ordered phases can be identified from their low-energy spectrum.

From the ordering patterns revealed in the sublattice magnetization histograms $(m_A, m_B)$ one can predict a particular combination of quantum numbers of a set of low-energy eigenstates in the symmetry-broken phases [52–54]. In the thermodynamic limit, these states become exactly degenerate and span the ground state manifold. On finite-size systems these states are expected to appear as the lowest-energy states and the finite energy gaps with reference to the finite-size ground state energy scale to zero exponentially with increasing system size $N$. For the Baxter FM phase one predicts the particular combination of the four states {$2 \times \Gamma$ even, $\Gamma$ odd, M odd} where $\Gamma = (0, 0)$ and M $= (\pi, \pi)$ denote the momentum $\mathbf{k}$ of the eigenstate, and even/odd denotes the eigenvalue under the global spin-inversion symmetry $\mathcal{S}$. For the $\langle\mu\rangle$ phase one expects another set of four states {$\Gamma$ even, $\Gamma$ odd, M even, M odd} as lowest energy states. The larger $\mathbb{Z}_2 \times \mathbb{Z}_2$ onsite symmetry group of the AT-TFI model, where the spins on both sublattices can be individually inverted, was not resolved in our simulations, but enforces the $\Gamma$ odd and M odd levels to form a degenerate set. In Fig. 6 we show the low-energy spectrum across the FM–$\langle\mu\rangle$ transition as a function of $J_{\mathrm{AT}}/J$ for a fixed $h/J = 1.2$ for two different system sizes $N = 16$ (left panel) and $N = 32$ (right panel). For small $J_{\mathrm{AT}}/J$ the four lowest energy states are precisely those predicted for the FM phase with a large gap to the next excitations. For larger $J_{\mathrm{AT}}/J \approx 0.875 - 1.5$ the second $\Gamma$ even level is shifted to higher energy while a M even level becomes a low-energy level such that the four lowest states are precisely the ones predicted for the $\langle\mu\rangle$ phase. The transition happens around $J_{\mathrm{AT}}/J \approx 0.875$, where the $\Gamma$ even and M even levels cross. For $J_{\mathrm{AT}}/J \gtrsim 1.5$ we observe that many levels in the energy spectrum start collapsing to the ground state rapidly, destroying the $\langle\mu\rangle$ phase. A further analysis of this strongly frustrated coupling regime is left for future studies.

Comparing the $N = 16$ and $N = 32$ systems in the FM and $\langle\mu\rangle$ phases, one clearly observes that the four expected ground state levels have approximately constant scaled energy gaps $\Delta_i^N \times N$ in the vicinity of the phase transition point, which means that the bare energy gaps scale to zero approximately as $1/N$. Further away from the phase transition, this collapse is even faster and eventually becomes exponential. The scaled energy gaps to the higher levels, on the other hand, scale upwards in $\Delta_i^N \times N$ such that their bare energy gaps are expected to be non-zero in the thermodynamic limit $N \to \infty$, so that indeed the expected four-fold degenerate ground state manifold is built.

The energy spectrum at the phase transition point between the FM and $\langle\mu\rangle$ phases is interesting by itself. It resembles an O(2) rotor spectrum in the ordered (Goldstone) phase where all levels above the unique ground state are two-fold degenerate and the gaps $\Delta_i^N \propto i^2/N$ [55], where we here, for simplicity, identify the two-fold (quasi-)degenerate levels with the same value of $i$. In contrast to a critical point (with dynamical critical exponent $z = 1$), the scaling of the energy gaps with system size here is inversely proportional to $N$ instead of $L = \sqrt{N}$. Such a rotor spectrum is characteristic for an O(2)-symmetry broken Goldstone phase [52–54]. The spectra shown in Fig. 6 obey these unique properties close to the phase transition point (indicated by a vertical dashed line at $J_{\mathrm{AT}}/J \approx 0.875$) where the low-energy gaps are found at the values $\Delta_i^N \times N \approx \{0, 1, 4, 9, \dots\}$, and all levels above the ground state are two-fold (quasi-)degenerate.

We want to emphasize, that we expect this rotor spectrum to be only approximate: At the transition point the $q = 4$ monopole operator in an effective field theory description vanishes. Away from the transition point this operator is non-zero and relevant and, depending on its sign, stabilizes directly either the FM or the $\langle\mu\rangle$ phases. At the transition, only the higher-order $q = 8$ monopole operator remains non-zero [see also the eight peak structure of the sublattice magnetization histogram in Fig. 3(d), center plot] and is relevant to eventually (as a function of system size) destroy the Goldstone phase. Due to its high order, the breakdown of the O(2) rotor spectrum will only be observed for larger system sizes. This somewhat surprising structure at this (weakly) first order transition is comparable to the situation reported recently

in quantum dimer and loop models on the square lattice [56–58], where an approximate Goldstone energy spectrum is also observed at a weakly first order transition. In contrast to those works, the weakly first order line we find, terminates at one point in a genuine 3D XY critical line, where the relevant $q = 4$ monopole operator of the Goldstone phase transmutes to the dangerously irrelevant $q = 4$ monopole perturbation of the 3D XY fixed point. It is quite rewarding that the study of the torus spectrum allows to identify these operators and scenarios using relatively modest system sizes in ED.

### 3.3 Connection to classical 3D Ashkin-Teller model

The 2D quantum-mechanical AT-TFI model can be effectively mapped onto a $2 + 1$D classical model by trotterization, where the additional dimension corresponds to imaginary time [see App. B for details]. The resulting classical model is anisotropic and thus microscopically rather different to the standard classical 3D Ashkin Teller (CAT) model. In the following we will, nevertheless, argue that both models will be equivalent in the scaling limit (in the here considered parameter regime).

The phase diagram obtained in this paper for the $2 + 1$D AT-TFI model is similar to parts of the phase diagram of the CAT model [31–36, 59]. In particular, the phases we obtain here are also found in the phase diagram of the CAT model in a similar arrangement. Also, the pattern of (weakly) first order and continuous phase transitions are identical in both models. Especially, the transition between the Baxter FM and the PM phases are also found to be either (weakly) first order or continuous in the CAT model, depending on the sign of the four-spin Ashkin-Teller coupling. However, the universality class of a line of continuous phase transitions in the CAT model from the PM to the Baxter FM and the $\langle \sigma \rangle$ (corresponding to the $\langle \mu \rangle$ phase in our language) phases has been a long debated and outstanding issue in the CAT model [32–36], where remnants of tricitical behaviour, extremely large crossover regions between continuous and first-order transitions [36], or even non-universal behaviour [34] have been suggested.

Based on universality arguments and the mentioned similarities of the phase diagrams, we expect that the results obtained in this paper for the AT-TFI model also apply to the critical behaviour in the CAT model. Therefore, we propose, that the transition from the PM to the Baxter FM and to the $\langle \sigma \rangle$ phases in the CAT model belong to the 3D XY universality class. Furthermore, we suggest that the critical point where all those three phases meet is not of tricritical nature, as previously suggested [33, 34], but also belongs to the 3D XY universality class. This point is still somewhat special, as it is the point along the XY transition line where the coupling constant of the dangerously irrelevant $q = 4$ monopole operator changes sign.

Eventually, the peculiar nature of the weakly first-order phase transition between the FM and the $\langle \mu \rangle$ phases might also be translated to the transition between the Baxter FM and the $\langle \sigma \rangle$ phases in the CAT model, explaining its weakly first order nature.

## 4 Analysis of the Toric Code Ising model

After we have provided the foundation with the extensive discussion of the topologically trivial, unconstrained AT-TFI model in the previous section we will now discuss the properties of the original TCI model, Eq. (1). We will particularly focus on the properties of the confinement-deconfinement transition between the $\mathbb{Z}_2$ TO and the $\mathbb{Z}_2$ SB phases.

### 4.1 Phase diagram

We show a sketch of the phase diagram for the TCI model in Fig. 7 together with a classification of the phases and types of phase transitions. The mapping between the TCI model and the AT-

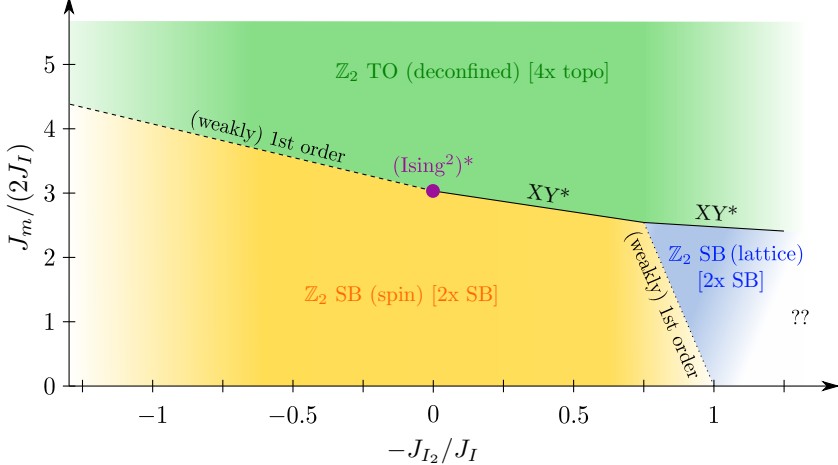

Figure 7: Sketch of the phase diagram for the TCI model, Eq. (1). Full lines denote continuous phase transitions, dashed lines (weakly) first order transitions. TO denotes topological order, $\mathbb{Z}_2$ SB denotes states that spontaneously break a $\mathbb{Z}_2$ symmetry, while (spin) indicates that the spin-inversion symmetry is broken, and (lattice) indicates that the sublattice exchange symmetry (part of the larger lattice space symmetry group) is broken. The question marks indicate regions we have not investigated in detail. The ground state degeneracy of the phases (on a torus) is given by the numbers in square brackets, where "topo" ("SB") indicates that the degeneracy is of topological (symmetry-broken) nature.

TFI model makes it possible to directly infer the phase diagram and types of phase transitions in the TCI model from the results of the unconstrained AT-TFI model. For that, we want to remind the reader that the $\mu^z$ operators are mapped to plaquette operators, $\mu_p^z = B_p$, and that the product of two $\mu^x$ operators on neighboring sites of the dual lattice maps to the $\sigma^x$ operator on the intermediate link of the original lattice, $\mu_p^x \mu_{p+\vec{x}(\vec{y})}^x = \sigma_{i(p)}^x$ [see App. A for details].

The PM phase of the AT-TFI model maps to the $\mathbb{Z}_2$ TO phase in the TCI model where the fractionalized anyonic excitations are deconfined [$\mathbb{Z}_2$ TO in Fig. 7]. The topological four-fold degenerate ground state (in the thermodynamic limit) is built from the lowest energy levels in all four different boundary condition sectors (periodic and antiperiodic around both directions of the torus) of the AT-TFI model, which become quasi-degenerate in this phase.

The FM (Baxter) phase in the AT-TFI model maps to the well-known Ising ferromagnet in the original $\sigma_i^x$ operators which spontaneously breaks the global $\mathbb{Z}_2$ spin rotational symmetry $R_z = \prod_i \sigma_i^z$ [$\mathbb{Z}_2$ SB (spin) in Fig. 7]: The constraint of globally even spin-inversion symmetry for the $\mu_i$ operators (symmetry operator $\mathcal{S}$) in the mapping from the AT-TFI to the TCI model deletes two of the four quasi-degenerate ground states of the (Baxter) FM phase, i.e. the $\Gamma$ odd and the M odd levels [see also Fig. 6]. The remaining two states, $\{2 \times \Gamma$ even (under $\mathcal{S})\}$, correspond to the two quasi-degenerate ground states of the standard Ising ferromagnet. They transform trivially under the sublattice exchange symmetry $\mathcal{S}_{A \leftrightarrow B}$, but non-trivially under the individual sublattice spin-inversion symmetries $\mathcal{S}_{A,B}$. Therefore, they are even and odd states under the spin-inversion operator for the $\sigma_i$ operators, $R_z$; their combination in the thermodynamic limit spontaneously breaks spin-inversion symmetry but not sublattice exchange symmetries (i.e. space symmetry group). The additional anti-periodic boundary condition levels have a gap which grows with linear system-size, $\Delta_i^N \propto L$, such that they can be ignored in the thermodynamic limit.

In the $\langle \mu \rangle$ phase the constraint of globally even spin inversion for the $\mu_i$ operators also

removes the two odd ($s_i = -1$) states of the four quasi-degenerate ground states. The two remaining states {$\Gamma$ even (under $\mathcal{S}$), M even (under $\mathcal{S}$)} now have different momentum quantum numbers [see also Fig. 6]. This pair of levels transforms trivially under the sublattice spin-inversion symmetries $\mathcal{S}_{A,B}$, but non-trivially under the sublattice exchange symmetry $\mathcal{S}_{A\leftrightarrow B}$ (note the non-zero momentum sector M). In the TCI model this corresponds to a state which spontaneously breaks a $\mathbb{Z}_2$ subgroup of the lattice space symmetry group (in particular translations $T_{x,y}$, mirror reflection $M_x$ and rotation $C_4$), but the $\sigma_i$ spin-inversion symmetry $R_z$ is, in contrast to the Ising ferromagnet, unbroken [$\mathbb{Z}_2$ SB (lattice) in Fig. 7]. Again, the anti-periodic boundary condition levels scale away with linear system size, $\Delta_i^N \propto L$, and do not influence the symmetry-broken ground state. A further discussion of the precise nature of this phase in terms of the $\sigma_i$ operators is left for future studies.

## 4.2 Spontaneous symmetry breaking enforced by symmetry fractionalization

The spontaneous breaking of spin rotation ($R_z$) and/or space group symmetries ($M_x, C_4, T_{x,y}$) across the confinement transitions is in fact constrained by the symmetry properties of the TCI model. More precisely, in the deconfined $\mathbb{Z}_2$ TO, the symmetry group $G_s$ in Eq. (4) is implemented projectively on the $m$ particles, whose condensation drives the confinement transition. The symmetry fractionalization class of the $m$ particles dictates the spontaneously broken symmetry in the confined phase, as we elaborate below.

The $\mathbb{Z}_2$ topological order in the TCI model features three types of anyons (or superselection sectors) in addition to local excitations (sector $1$): electric charge $e$, magnetic vortex $m$ and fermion $\epsilon = e \times m$. In the $J_e \to +\infty$ limit of interest here, electric charges $e$ and fermions $\epsilon$ are both absent, leaving $m$ the only low-energy excitations in the deconfined topological order. The symmetry group $G_s$ in Eq. (4) is implemented on the $m$ particles projectively, manifested by a nontrivial symmetry fractionalization class [5,27–30] [$\omega_m$] $\in \mathcal{H}^2(G_s, \mathbb{Z}_2)$ of the $m$ particles:

$$U_g^m U_h^m = \omega_m(g,h) U_{gh}^m, \quad \forall\ g,h \in G,\ \omega_m(g,h) = \pm 1,$$
$$\omega_m(g,h)\omega_m(gh,k) = \omega_m(g,hk)\omega_m(h,k), \tag{13}$$

where $U_g^m$ is the localized symmetry action [28–30] on a single $m$ particle, and [$\omega_m$] belongs to the 2nd group cohomology of $\mathbb{Z}_2$ coefficient $\omega_m(g,h) = \pm 1$.

In the deconfined phase of the TCI model, for $m$ particles, the gauge-invariant nontrivial elements $\omega_m(g,h) \neq 1$ have the following form:

$$U_{R_z M_x}^m U_{R_z M_x}^m = \omega_m(R_z M_x, R_z M_x) = -1; \tag{14}$$

$$U_{R_z}^m U_{T_\alpha}^m (U_{R_z}^m)^{-1}(U_{T_\alpha}^m)^{-1} = \frac{\omega_m(R_z, T_\alpha)}{\omega_m(T_\alpha, R_z)} = -1, \quad \alpha = x,y; \tag{15}$$

$$U_{R_z}^m U_{C_4}^m (U_{R_z}^m)^{-1}(U_{C_4}^m)^{-1} = \frac{\omega_m(R_z, C_4)}{\omega_m(C_4, R_z)} = -1. \tag{16}$$

In other words, the local symmetry actions {$U_g^m | g \in G_s$} form a projective representation of the symmetry group $G_s$, characterized by a nontrivial central extension $G_g$ of group $G_s$ with a $\mathbb{Z}_2$ center, as shown in Eq. (10). In fact, the above algebra between symmetry operations is in one-to-one correspondence with the symmetry relations Eqs. (6)-(8) in the dual AT-TFI model.

Increasing $J_I/J_m$ and $|J_{I_2}|/J_m$ condenses $m$ particles, driving the system into a confined phase. In such an anyon condensation transition [60–63] enriched by symmetry $G_s$, there is a general theorem [16] about the spontaneously broken symmetries across the continuous phase transition. In particular, in a $G_s$-symmetric topological order (or mathematically a unitary braided fusion category $\mathcal{C}$ [5]), if the symmetry group $G_s$ does not permute different types of anyons, the symmetry fractionalization class is classified by the 2nd group cohomology

$\mathcal{H}^2(G_s, \mathcal{A})$ valued in Abelian anyons $\mathcal{A}$ [28–30], such as $(\mathbb{Z}_2)^2 = \{1, e, m, \epsilon\}$ in the case of the Toric Code. If a continuous phase transition is driven by condensing the Abelian anyon $a$ (i.e. with condensable algebra [61] $A = 1 + a + a^2 + \cdots$), the following theorem holds [16]:

*Theorem: The symmetry $G_s$ is preserved across a continuous phase transition driven by condensing anyon $a$ if and only if the fractionalization class $[\omega_a(g, h)] \in \mathcal{H}^2(G_s, \mathcal{A})$ of anyon $a$ is trivial in the topological order $\mathcal{C}$ enriched by symmetry $G_s$.*

In other words, if the condensed anyon $a$ has a nontrivial symmetry fractionalization class, the resulting phase must spontaneously break the symmetry $G_s$ to a subgroup.

Applying the above theorem to our case of $m$-condensation transition out of the TO phase in the TCI model, based on the nontrivial symmetry fractionalization class summarized in Eqs. (14)-(16), it becomes clear that in the confined phase, either onsite spin rotational symmetry $R_z$ or the crystal symmetry $p4gm$ (generated by $M_x, T_{x,y}, C_4$) must be spontaneously broken. Translating into the language of the dual AT-TFI model: since the global Ising symmetry $\mathcal{S}$ must be broken across the phase transition, according to Eqs. (6)-(8), it is impossible to break the Ising symmetry $\mathcal{S} = (S_A M_x)^2$, if sublattice symmetry $S_A$ and crystal symmetry $M_x$ (similarly for translation $T_a$ and rotation $C_4$) are both preserved. As a result, there are two possible fates for the confined phase:

(1) Preserving the symmetry $R_z$ [$\mathcal{S}_{A(B)}$ on one sublattice in the dual model], but spontaneously breaking the spatial symmetry $G_s = p4gm$, realized in the $\mathbb{Z}_2$ SB (lattice) phase of the TCI model [$\langle \mu \rangle$ phase in the dual model];

(2) Preserving the spatial symmetry $p4gm$, but spontaneously breaking the $R_z$ spin inversion symmetry [$\mathcal{S}_{A,B}$ symmetry on both sublattices in the dual model], realized in the $\mathbb{Z}_2$ SB (spin) phase of the TCI model [Baxter FM phase in the dual model].

Clearly, the spontaneous SB patterns enforced by symmetry fractionalization in the Toric Code is in full agreement with the phase diagrams of the TCI and AT-TFI models computed in this paper [see Fig. 7 and Fig. 2].

Finally we comment on the implications of the nontrivial fractionalization class to emergent symmetries of the $m$-condensation transition. As an example we consider the mirror symmetry fractionalization summarized in Eq. (14). As a consequence of the central extension, Eq. (10), the original $G_s = \mathbb{Z}_2$ symmetry group generated by the mirror symmetry $M_x$ is enlarged to a $G_g = \mathbb{Z}_4$ group generated by $M_x S_A$, due to the $\mathbb{Z}_2$ gauge structure in the Toric Code. In the LGW Hamiltonian Eq. (11) in the dual language, the two order parameter fields $(\phi_1, \phi_2)$ transform as

$$\psi \equiv \phi_1 + i\phi_2 \xrightarrow{M_x S_A} -\phi_2 + i\phi_1 = i\psi. \qquad (17)$$

At the stable $O(2)$ fixed point of the LGW Hamiltonian, when $\psi^4 + h.c.$ and $|\psi|^4$ terms become irrelevant, the critical point acquires an enlarged emergent $O(2) \simeq U(1)$ symmetry $\psi \to e^{i\theta} \psi$. This leads to the emergent $XY^*$ critical line in the phase diagram, Fig. 7. We note that the emergence of an enlarged infrared symmetry which is larger than the ultraviolet symmetry in the microscopic lattice model, is a ubiquitous phenomenon due to the diverging correlation length at a critical point or in a critical phase. The deconfined quantum critical points [1] and the Stiefel liquids [64] are candidate examples of conformally invariant fixed points of such nature.

## 4.3 Phase transitions

The fractionalized excitations in the $\mathbb{Z}_2$ TO phase undergo the corresponding conventional transitions observed for the AT-TFI model – i.e. first order, Ising$^2$, and XY – when they condense and become confined by the increased loop tension through the Ising interactions in the symmetry broken phases. The critical theories are then the *starred* versions of the classical theories where the condensing particles are not fractionalized [4, 17, 18]. We thus observe a line

of first-order transitions between confined and deconfined phases, a fine-tuned 2+1D *(Ising²)\** and, most interestingly, a line of emergent 2+1D *XY\** transitions [4,14,65] between the $\mathbb{Z}_2$ TO phase and the $\mathbb{Z}_2$ SB phases. The first-order and XY* transitions are induced by the coupling among magnetic vortices, i.e. the $m$ anyons, on the two sublattices of the dual lattice, such that we expect those to be generic for a $\mathbb{Z}_2$ TO to $\mathbb{Z}_2$ SB phase transition, independent of the microscopic model, while the (Ising²)* transition can only be reached by fine-tuning [26]. The transition between the two distinct SB phases does not feature any deconfined quasi-particles and, thus, remains a standard (weakly) first order transition.

In the following, we will discuss the unconventional critical points in more detail. We will compute their distinct CTESs and show that they feature unusually large critical exponents for the spatial decay of the correlations, $\eta^*$.

### 4.3.1 (Ising²)* transition

Once more, let us first consider the special case of vanishing next-to-nearest neighbor Ising interactions $J_{I_2} = 0$. This corresponds to a vanishing 4-spin interaction $J_{AT} = 0$ in the AT-TFI model where the two sublattices decouple. The fractional $m$ particles then undergo the Ising² transition, such that we call this transition (Ising²)*. Although the (Ising²)* transition can only be reached by fine-tuning discussing it is very illustrative since many of its properties can be exactly inferred from the standard Ising transition.

We show the CTES for the (Ising²)* critical point in Fig. 8(a), together with the Ising² CTES as a comparison. The (Ising²)* CTES is obtained from the Ising² CTES by removing all odd levels under the global spin-inversion symmetry $\mathcal{S}$ (empty circles), and including levels, which correspond to the other topological ground state sectors in the $\mathbb{Z}_2$ TO phase. We compute these levels from finite-size simulations of the AT-TFI model using anti-periodic boundary conditions along one or both directions of the torus [denoted A/P, A/A in Fig. 8, respectively] and extrapolate the results to the thermodynamic limit as done for the periodic levels. The resulting (Ising²)* CTES is very characteristic and features low-lying levels in the topologically non-trivial sectors. This is similar to what has been observed in the CTES for the Ising* transition between a $\mathbb{Z}_2$ TO and a paramagnetic, confined phase [17]. We expect this feature to be characteristic for *starred* critical points where deconfined excitations condense. The gaps to the non-trivial topological sectors A/P, A/A are, however, twice as large as in the Ising* CTES (up to the accuracy of the extrapolations). This can again be understood by the decoupling of the sublattices, such that levels in the Ising² spectrum can be constructed from pairs of levels in the standard Ising spectrum, which applies individually to each boundary condition sector.

Apart from the CTES, another very characteristic feature of confinement-deconfinement transitions, where the confined phase is symmetry broken and has a finite order-parameter, is an unusually large critical exponent $\eta^* \gg \eta$ [11–14] which describes the decay of the order parameter correlations at criticality

$$C(r) = \langle \sigma_0^x \sigma_r^x \rangle \propto r^{-D+2-\eta^*}. \tag{18}$$

Starred critical exponents denote the values for the confinement transition while the unstarred critical exponents denote the values for the corresponding classical transition, and $D = 3$. The correlation length exponent, in contrast, is identical to the one from the corresponding classical transition for the fractionalized field, $\nu^* = \nu$ [11–14].

In the fine-tuned (Ising²)* case discussed here, $\eta^*$ can be calculated explicitly from the standard value of $\eta$ for the Ising transition, as already described in Ref. [26]. Starting within the deconfined phase, applying a single $\sigma_i^x$ operator on the ground state creates two fractional, deconfined excitations on the plaquettes sharing site $i$, i.e. magnetic vortices with $B_p = -1$, which are precisely the $m$ particles. These fractional excitations live, in the fine-tuned $J_{I_2} = 0$

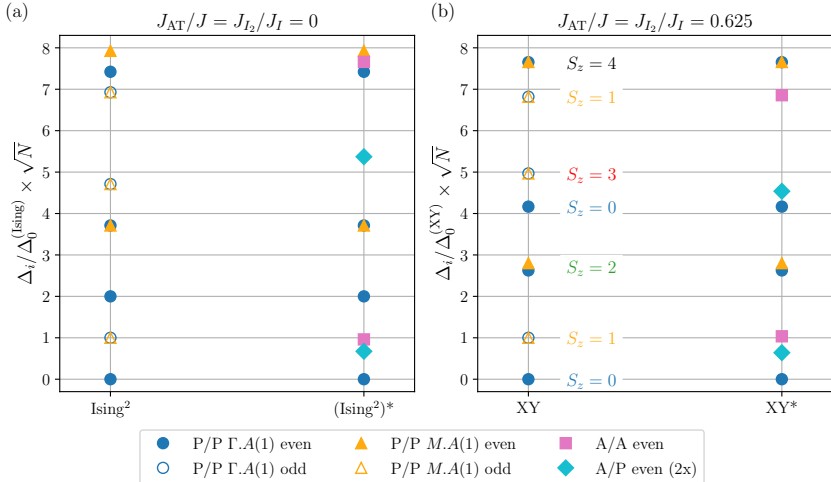

Figure 8: CTES for the (a) (Ising$^2$)*, (b) XY* transitions for $\kappa = 0$. As a comparison, we show the CTES for the conventional Ising$^2$ and XY universality classes in the $\kappa = 0$ sector. The labels A/P etc. denote the boundary conditions along the two directions of the torus, where P (A) means periodic (antiperiodic). The boundary conditions correspond to the four different topological sectors of the $\mathbb{Z}_2$ TO phase on the torus. Full (empty) symbols denote levels that are even (odd) under the global spin-inversion symmetry $\mathcal{S}$ of the $\mu_i$ operators. In (b) we also indicate the emergent $S_z$ sector of the spectral levels.

case, on the two distinct sublattices and can, thus, be treated independently. The correlation function $C(r)$, therefore, exactly fractionalizes in terms of the $\mu$ operators on the dual lattice

$$\begin{aligned} C(r) &= \langle \sigma_0^x \sigma_r^x \rangle = \langle \mu_{0,A}^x \mu_{0,B}^x \mu_{r,A}^x \mu_{r,B}^x \rangle \\ &= \langle \mu_{0,A}^x \mu_{r,A}^x \rangle \langle \mu_{0,B}^x \mu_{r,B}^x \rangle. \end{aligned} \tag{19}$$

The last relation follows from the decoupling of the sublattices of the dual lattice when $J_{I_2} = 0$ (i.e. $J_{\mathrm{AT}} = 0$). From Eqs. (18, 19), the critical exponent $\eta^*$ ($\sigma$ operators) can be explicitly computed from $\eta$ ($\mu$ operators)

$$\begin{aligned} -D + 2 - \eta^* &= 2(-D + 2 - \eta) \\ \Rightarrow \eta^* &= D - 2 + 2\eta. \end{aligned} \tag{20}$$

For the case discussed here, $D = 3$, $\eta^* = 1 + 2\eta$, and with $\eta \approx 0.036$ in for the Ising transition [51] we compute the unusually large characteristic $\eta^* \approx 1.072$ for (Ising$^2$)*. For this particular transition we can thus understand the unusually large value for the critical exponent $\eta^*$ on a microscopic level. The (Ising$^2$)* transition is therefore a pedagogically very interesting example for confinement-deconfinement transitions in general.

In Fig. 9(a) we show results for the correlation function $C(r = L/4)$ obtained in terms of the AT-TFI model via Eq. (19) around the critical point for different system sizes of linear size $L$, rescaled by a standard finite-size scaling ansatz. We use the literature value for $\nu$ for the Ising transition [see also Fig. 14(a) in the Appendix] and then tune $\eta^*$ such that we observe a good collapse of $C(L/4)$. As expected, the obtained $\eta^* = 1.01(3)$ is much larger than the value for a standard Ising transition $\eta \approx 0.036$ due to the fractionalization of the quasi-particles and is close to the predicted value $\eta^* \approx 1.072$.

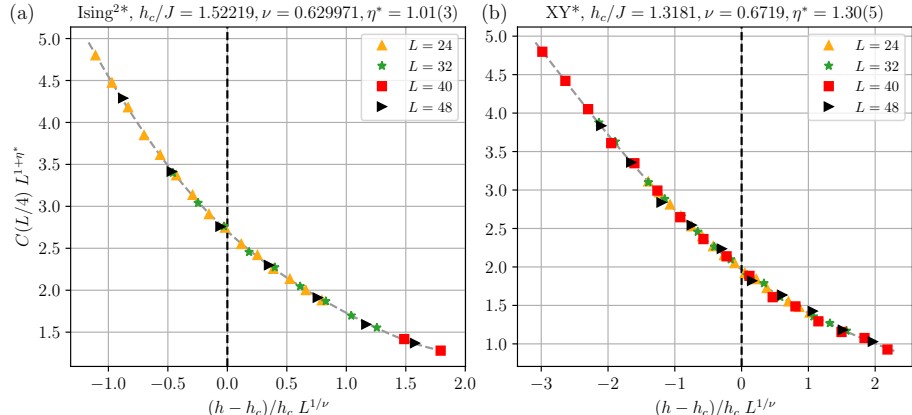

Figure 9: Collapse plots for correlation function $C(L/4)$ across the $(\text{Ising}^2)^*$ transition (a), and the XY* transition for $J_{\text{AT}}/J = 0.625$ (b). The dashed, grey line shows a polynomial fit of the data as a guide to the eye. We use the known critical exponents $\nu$ for the Ising and XY transitions [51] in (a), (b), respectively. The critical exponents $\eta^*$ are chosen such that the best collapse of $C(L/4)$ is observed (minimal root-mean-square distance of the data to the fit). The obtained values for $\eta^*$ are much larger than the standard Ising or XY values due to the fractionalization of the quasiparticles. See text for further details.

### 4.3.2 XY* transition

As above, we also construct the CTES for the XY* transition, which is shown in Fig. 8(b). The XY* CTES is qualitatively different from the Ising* and the $(\text{Ising}^2)^*$ CTESs such that they can be easily distinguished. However, the lowest levels are again built from the four different topological sectors, supporting once more our assumption that this is a general feature for *starred* transitions where fractional particles condense.

Also the XY* transition can be characterized by an unconventionally large critical exponent $\eta^*$ for the scaling of the correlation function while the exponent for the scaling of the correlation length, $\nu^*$, remains the classical XY value $\nu^* = \nu$, since the two-particle correlation function $\langle \sigma_0^x \sigma_r^x \rangle$ effectively becomes a four-particle correlation function of the fractionalized particles [14]. For the XY* transition, $\eta^*$ cannot be computed exactly from $\eta$ because the fractionalized particles are coupled. In Fig. 9(b) we show a collapse plot of $C(r = L/4)$ for lattices with different linear sizes $L$ across the XY* transition from which we obtain $\eta^* \approx 1.30(5)$. This is much larger than the classical value $\eta \approx 0.04$ [51], demonstrating the fractionalization of the condensing particles. The observed value for $\eta^*$ is lower than the expected value for a composite operator in the XY theory, $\eta^* \approx 1.47$ [4, 14, 66]. This discrepancy might be due to the emergent nature of the XY* transition which could introduce stronger finite-size corrections and make it more difficult to obtain the critical exponent precisely.

## 5 Conclusion

We have investigated the properties of a confinement-deconfinement phase transition between a $\mathbb{Z}_2$ TO phase and a $\mathbb{Z}_2$ SB phase using the prime example of the Toric Code perturbed by Ising interactions on nearest and next-to-nearest neighbors. We have used the modern approach of measuring the CTES using ED combined with QMC simulations to identify the type of phase transitions between the different phases. We obtain three distinct types of transitions between the topologically ordered and the $\mathbb{Z}_2$ SB phases: An $(\text{Ising}^2)^*$ transition for vanishing second-

neighbor Ising interactions, a line of (weakly) first-order transitions, and most interestingly, a line of emergent XY* transitions. The $(Ising^2)$* transition can only be reached by fine-tuning, therefore we expect that a generic transition between a $\mathbb{Z}_2$ TO and a $\mathbb{Z}_2$ SB phase is either described by a (weakly) first-order transition or a XY* critical point, independent of the specific microscopic model. It is fascinating that the $\mathbb{Z}_2$ symmetry breaking phase transition is *not* in the Ising universality class as one would naively expect from a Ginzburg Landau symmetry analysis based on the nature of the order parameter. Instead, as we explained in this work, the adjacent $\mathbb{Z}_2$ topological ordered phase with its fractional excitations promotes the phase transition a XY* universality class. We believe that the TCI model studied here is one of the simplest and prototypical instances of a beyond LGW transition involving spontaneous symmetry breaking.

Along the way, we have mapped the perturbed Toric Code model to a transverse field Ising model with additional Ashkin-Teller like four-spin interactions, which we used to perform extensive numerical simulations. In the latter, topologically trivial, model we were able to identify the emergent XY critical points by comparing the measured CTES to the known CTES for XY universality. This is a highly non-trivial test and demonstrates the power of critical torus energy spectroscopy, already on systems of only a few ten spins.

Furthermore, we have constructed the universal CTESs for the $(Ising^2)$* and XY* critical points which feature a universal fingerprint for the corresponding universality classes in $D = 2 + 1$ dimensions. These further extend the catalogue of already charted CTESs and can be used as a reference to identify these universality classes in the future.

In a more standard approach, we have also estimated the critical exponents $\nu$ and $\eta^*$ for the $(Ising^2)$* and the XY* transitions. The latter obtains a strongly increased value compared to the Ising and XY transitions because of the fractionalized particles which condense at the critical points. In the case of the $(Ising^2)$* transition, the value of $\eta^*$ can be computed directly from the exponent $\eta$ of the standard Ising transition, making this transition particularly interesting in a pedagogical sense.

We have also computed the non-trivial symmetry fractionalization class of the condensing anyons with respect to the global $\mathbb{Z}_2$ symmetry and the lattice space symmetry group. This analysis implies that a condensed phase has to be symmetry-broken and, in particular, either break the $\mathbb{Z}_2$ spin-inversion or the space group symmetry of the TCI model. Both of these predicted symmetry breaking patterns are observed in the numerically obtained phase diagram and can be identified using energy level spectroscopy.

Our findings for the topologically trivial, mapped model are in agreement with the phase transitions one expects in a $n = 2$-component scalar field LGW theory with cubic anisotropy. The same field theory also describes the original topological model, where the fields describe the fractionalized particles and thus have to fulfill additional constraints, giving rise to the *starred* critical points. This demonstrates, that many of our results are not specific to the chosen microscopic model, but hold generally for transitions between $\mathbb{Z}_2$ TO to $\mathbb{Z}_2$ SB phases.

The results presented in this paper demonstrate that torus energy spectroscopy can be readily applied to identify and characterize non-trivial critical behaviour. It will be fruitful to study other models featuring quantum spin liquid phases to chart and inspect other possible deconfined quantum critical points, such as the more general O(N)* ones, or deconfined criticality in designer Hamiltonians [2], in future studies.

# Acknowledgements

We thank S. Whitsitt and S. Sachdev for earlier collaborations on related problems. YML and AML thank the hospitality of KITP.

**Funding information** MS and AML acknowledge support by the Austrian Science Fund through project I-4548. YML acknowledges support by the National Science Foundation under Grant No. NSF DMR-1653769. YML and AML thank the hospitality of KITP, where this work was supported in part by the National Science Foundation under Grant No. NSF PHY-1748958. The computational results presented have been achieved in part using the Vienna Scientific Cluster (VSC) and in part using the LEO HPC infrastructure of the University of Innsbruck.

# A   Mapping Toric Code models to transverse field Ising models

In this appendix, we demonstrate the exact mapping of the charge-free sector of the TCI model to the AT-TFI model. A similar mapping has been used in previous studies of the Toric Code in a magnetic field [25,37,38] and, in particular, by the authors of this paper in Ref. [17] (supplemental material), where the additional constraints of globally even spin-inversion symmetry and different boundary condition sectors have been worked out in detail. The mapping from the TCI to the AT-TFI model follows the same steps as shown in the supplemental material of Ref. [17]. For the benefit of the readers, we will reproduce the details of the mapping in this appendix, and apply it to the TCI model.

The Hamiltonian of the TCI model, Eq. (1), we want to consider here is given by

$$
\begin{aligned}
H = & -J_e \sum_s A_s - J_m \sum_p B_p \\
& -J_I \sum_{\langle i,j \rangle} \sigma_i^x \sigma_j^x - J_{I_2} \sum_{\langle\langle i,j \rangle\rangle} \sigma_i^x \sigma_j^x - J'_{I_2} \sum_{\langle\langle i,j \rangle\rangle'} \sigma_i^x \sigma_j^x,
\end{aligned}
\tag{A.1}
$$

with the star and plaquette operators given by $A_s = \prod_{i \in s} \sigma_i^x$ and $B_p = \prod_{i \in p} \sigma_i^z$, respectively. Here, the $\sigma_i$ describe spins on the links of a square lattice, $p$ denotes a plaquette and $s$ a star on this lattice [see Fig. 10]. We will only consider the case $J_e, J_m \geq 0$. $J_I$ denotes the Ising coupling between nearest neighbour sites, $J_{I_2}$ is the coupling between next-to-nearest neighbour Ising interactions along the edges of the lattice, while $J'_{I_2}$ denotes the coupling between next-to-nearest neighbour interactions among sites on opposite edges of the plaquettes.

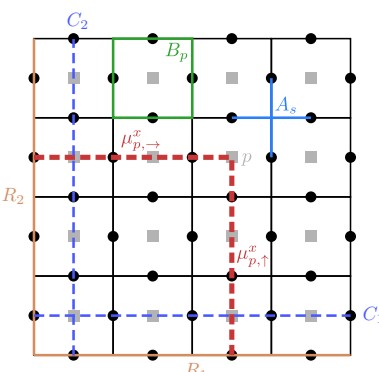

Figure 10: Toric Code on a torus. Black dots show the positions of the Toric Code variables $\sigma_i^{x,z}$, grey squares the dual lattice for the variables $\mu_p^{x,z}$. $C_{1,2}$ depict an (arbitrary) choice of the two incontractible loops winding around the torus, which define the Wilson loop operators $t_{1,2}$. The reference lines $R_{1,2}$ are used to define the reference positions for the operators $\mu_{p,\rightarrow(\uparrow)}^x$. See text for further details.

Let us first recall some properties of the unperturbed Toric Code [22]. All $A_s$ and $B_p$ com-

mute with each other such that the GS of $H$, for $J_I = J_{I_2} = J'_{I_2} = 0$, can be found by setting $A_s = 1 \, \forall s$ and $B_p = 1 \, \forall p$. On a torus, however, not all of the $A_s$ and $B_p$ are linearly independent, as

$$\prod_s A_s = 1, \text{ and } \prod_p B_p = 1. \tag{A.2}$$

These constraints lead to a four-fold degenerate ground-state manifold on a torus. The ground states can be distinguished by the eigenvalues $\pm 1$ of two Wilson loop operators

$$t_{1,2} = \prod_{i \in C_{1,2}} \sigma_i^x, \tag{A.3}$$

where the closed paths $C_{1,2}$ wind around the torus along the two different non-contractible loops, as illustrated in Fig. 10. The precise shape of the paths $C_{1,2}$ is arbitrary up to local deformations; they can be modified by adding any contractible closed loop of $\sigma_i^x$ operators. Such a contractible loop can be written as the product of the enclosed $A_s$ operators. In the ground-state manifold, all $A_s$ have eigenvalues $+1$, therefore changing the paths $C_{1,2}$ locally does not alter the eigenvalues of the $t_{1,2}$ operators.

To perform the mapping from the TCI to the AT-TFI model we first note, that the operators $A_s$ and $t_{1,2}$ are still conserved for non-zero Ising interactions. So, we will only consider the charge-free sector, $A_s = 1 \, \forall s$, and set $J_e \gg J_m$ such that the low-energy physics is free of charge excitations ($A_s = -1$). On each site $p$ of the dual lattice (center of plaquette $p$) we define the new variables [37]

$$\mu_p^z = B_p, \tag{A.4}$$

$$\mu_{p,\to(\uparrow)}^x = \prod_{i \in c_{p \to (\uparrow)}} \sigma_i^x. \tag{A.5}$$

To define the paths $c_{p \to (\uparrow)}$, we choose two incontractible reference paths $R_{1,2}$ in $\hat{x}(\hat{y})$ direction along the lattice; $c_{p \to (\uparrow)}$ is then a straight path from $R_{2(1)}$ to the site $p$ in $\hat{x}(\hat{y})$-direction along the dual lattice [see Fig. 10]. It is straightforward to show that these new variables fulfill the Pauli algebra for spin operators, $\{\mu_p^x, \mu_p^z\} = 0, (\mu_p^x)^2 = 1, (\mu_p^z)^2 = 1$, and that

$$\sigma_i^x(\hat{x}) = \mu_{p(i),\uparrow}^x \mu_{p(i)-\hat{y},\uparrow}^x, \tag{A.6}$$

$$\sigma_i^x(\hat{y}) = \mu_{p(i),\to}^x \mu_{p(i)-\hat{x},\to}^x, \tag{A.7}$$

where $\sigma_i^x(\hat{x}(\hat{y}))$ describes a Pauli operator on a link in $\hat{x}(\hat{y})$-direction on the lattice.

With these definitions, the TCI model can be mapped to the AT-TFI model

$$H_{\text{AT}} = -h \sum_i \mu_i^z - J \sum_{\langle\langle i,j \rangle\rangle} \mu_i^x \mu_j^x - J' \sum_{\langle\langle\langle i,j \rangle\rangle\rangle} \mu_i^x \mu_j^x$$
$$+ J_{\text{AT}} \sum_i \mu_i^x \mu_{i+\hat{x}}^x \mu_{i+\hat{y}}^x \mu_{i+\hat{x}+\hat{y}}^x \tag{A.8}$$

on the dual lattice and $A_s = 1 \, \forall s$, as it was imposed. The couplings are given in terms of the original couplings as $h = J_m$, $J = 2J_I$, $J' = J'_{I_2}$ and $J_{\text{AT}} = -2J_{I_2}$. Figure 11 depicts an illustration how the individual interactions in the TCI model are mapped to the interactions in the AT-TFI model.

For simplicity, we set the coupling $J'_{I_2} = J' = 0$ in the main text, and only consider next-to-nearest neighbor Ising interactions along the edges (with coupling constant $J_{I_2}$) of the original TCI model, Eq. (1). While finite $J'$ alters the precise shape of the phase boundaries, we do not expect it to qualitatively change the nature of the phase transitions (as long as $J'$ is not too large), since it also preserves the two-sublattice structure of Eq. (2).

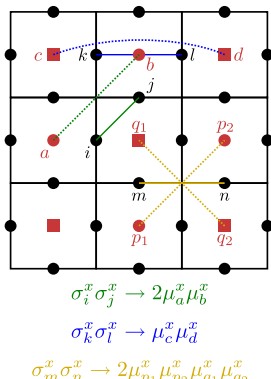

$$\sigma_i^x \sigma_j^x \rightarrow 2\mu_a^x \mu_b^x$$
$$\sigma_k^x \sigma_l^x \rightarrow \mu_c^x \mu_d^x$$
$$\sigma_m^x \sigma_n^x \rightarrow 2\mu_{p_1}^x \mu_{p_2}^x \mu_{q_1}^x \mu_{q_2}^x$$

Figure 11: Mapping of the Ising interactions from the TCI to the AT-TFI model. Solid lines show the original interactions, dotted lines (in the same color) the mapped interactions. The two sublattices of the dual square lattice are denoted by red squares/circles. The resulting two-body Ising interactions do not couple the sublattices, while the four-body interaction couples them.

The resulting AT-TFI model Eq. (A.8) is invariant under global spin-inversion $\mathcal{S} = \prod_p \mu_p^z$. From Eq. (A.4) and Eq. (A.2) it immediately follows that

$$\mathcal{S} = \prod_p B_p = 1. \tag{A.9}$$

The last equality is always satisfied on a torus and so the TCI model maps to an *even* AT-TFI model, where only states which are even representations of $\mathcal{S}$ are allowed.

Let us eventually consider the mapping of the different ground state sectors characterized by the eigenvalues of $t_{1,2}$. Using Eq. (A.6) and Eq. (A.7) it follows that

$$t_1 = \prod_{p=0}^{L-1} \mu_{(p,j)}^x \mu_{(p+1,j)}^x = \mu_{(0,j)}^x \mu_{(L,j)}^x, \tag{A.10}$$

where the index $(p, j)$ labels the position $p\hat{x} + j\hat{y}$ on the dual lattice and $L$ is the linear extent of the torus. Since $\mu_{(L,j)}^x$ is the periodic image of $\mu_{(0,j)}^x$ (along the $\hat{x}$-direction of the torus) and the eigenvalues of $t_1$ are $\pm 1$, we need to consider both periodic and anti-periodic boundary conditions for the $\mu$ operators on the dual lattice. An equivalent relation can be computed for $t_2$ along the $\hat{y}$-direction of the torus. The different ground state sectors of the Toric Code, therefore, map onto the four combinations of periodic and anti-periodic boundary conditions of the AT-TFI model for both directions around the torus.

Finally, we consider the $\mathbb{Z}_2$ symmetry operation $R_z = \prod_i \sigma_i^z$ of the original Hamiltonian Eq. (A.1), which is spontaneously broken in the $\mathbb{Z}_2$ spin symmetry-broken phase (see main text). It is easy to see that this symmetry operation is the product of the $B_p$ plaquette operators on every second plaquette (in a staggered arrangement). Therefore, using Eq. (A.4), we obtain

$$R_z = \prod_{p \in \text{SL } A} \mu_p^z = \prod_{p \in \text{SL } B} \mu_p^z, \tag{A.11}$$

where the product is over all plaquettes corresponding to either of the two sublattices $A$ or $B$ of the dual lattice.

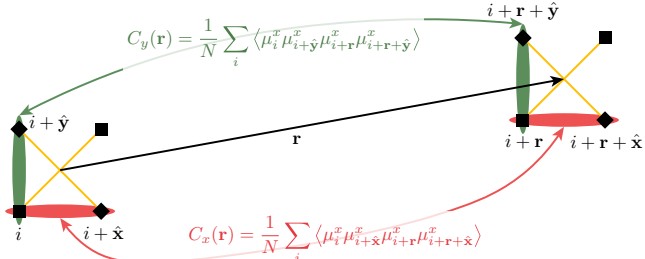

Figure 12: The correlation function $C_a(\mathbf{r})$ is taken between parallel pairs of neighboring sites at a distance $\mathbf{r}$. It can be either between horizontal pairs ($a = \hat{x}$, in red) or vertical pairs ($a = \hat{y}$, in green).

## B  Details about the quantum Monte Carlo simulations for the AT-TFI model

The QMC simulations were obtained via discrete-time world-line Monte Carlo. The original 2D quantum model Eq. (2) (at $J' = 0$) is mapped onto a 3D classical model by trotterization. This model is obtained by introducing $M$ imaginary-time slices, so that

$$\exp\left(-\beta H_{\text{AT}}\right) \approx \left[\exp\left(\frac{\beta h}{M}\sum_i \mu_i^z\right)\right. \tag{B.1}$$
$$\left.\times \exp\left(-\frac{\beta J}{M}\sum_i \mu_i^x \mu_{i+\hat{x}+\hat{y}}^x \left(1 - \rho\,\mu_{i+\hat{x}}^x \mu_{i+\hat{y}}^x\right)\right)\right]^M,$$

with $\rho = J_{\text{AT}}/J$ and $\beta = 1/T$ the inverse temperature.

By introducing $M$ identities (in the $\mu_i^x$ basis), the $\mu_i^z$ part turns into an Ising coupling $\lambda$ in the imaginary-time dimension. The resulting model is a 3D classical model with variables $s_\alpha = \pm 1$, on an $L \times L \times M$ lattice :

$$H_{\text{3D}} = -\lambda \sum_\alpha s_\alpha s_{\alpha+\hat{z}} \tag{B.2}$$
$$-\tilde{J}\sum_\alpha s_\alpha s_{\alpha+\hat{x}+\hat{y}}\left(1 - \rho\,s_{\alpha+\hat{x}}s_{\alpha+\hat{y}}\right),$$

where $\tilde{J} = J/M$ and $\lambda = -\log\tanh(h\beta/M)/(2\beta)$. The vector $\hat{z}$ connects consecutive imaginary-time slices of the effective model $H_{\text{3D}}$.

Additionally to the standard local Metropolis update, we used a variation of the Wolff cluster update [67]. Each cluster starts at a random site, and only grows on a single sublattice. When growing a cluster on sublattice $A$ ($B$), all spins on sublattice $B$ ($A$) are frozen. Therefore, the growth of the cluster is done on an effective pure Ising model where the couplings along each diagonal of a plaquette is decorated by the values of the other two spins of the same plaquette

$$J_{i,i+\hat{x}+\hat{y}}^{\text{eff}} \leftarrow \tilde{J}\left(1 - \rho\,s_{\alpha+\hat{x}}s_{\alpha+\hat{y}}\right). \tag{B.3}$$

Each update is then guaranteed to satisfy detailed balance (this has been checked against simulations using only single spin-flip updates). If $|\rho| < 1$, the effective coupling is always positive, ensuring the efficiency of the algorithm. For larger values of $|\rho|$, the algorithm may lose its efficiency, but this was not observed in our computations in the considered parameter regime.

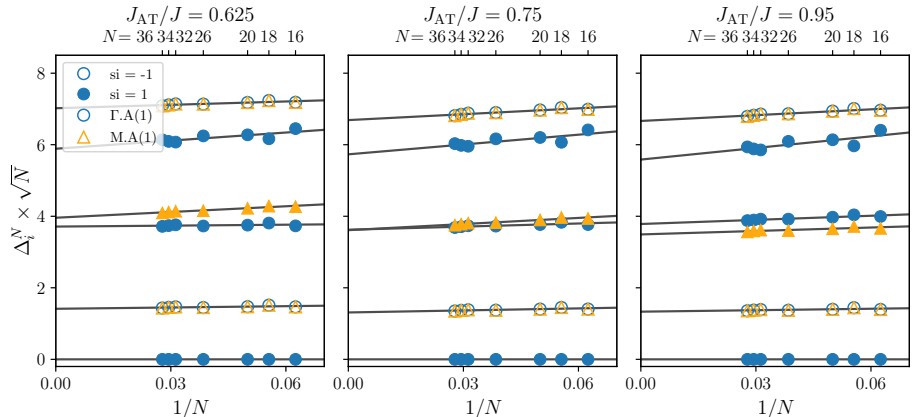

Figure 13: Finite-size extrapolations of the scaled energy gaps of the AT-TFI model at $J_{AT}/J > 0$. We perform linear expansions of the scaled finite-size energy gaps $\Delta_i^N \times \sqrt{N}$ (symbols) in $1/N$ (grey lines) to obtain the corresponding CTESs for $N \to \infty$ (i.e. $1/N = 0$). The CTESs are also shown in Fig. 5 in the main text and identified to belong to the XY/O(2) universality class.

A similar algorithm was recently introduced to study the 3D Ashkin-Teller model in Ref. [59]. We estimated the four-point correlation function between two parallel pairs of neighboring sites (as described in Fig. 12):

$$C_a(\mathbf{r}) = \frac{1}{N} \sum_i \langle \mu_i^x \mu_{i+\hat{a}}^x \mu_{i+\mathbf{r}}^x \mu_{i+\mathbf{r}+\hat{a}}^x \rangle_{GS} \tag{B.4}$$

$$\equiv \frac{1}{NM} \sum_\alpha \langle s_\alpha s_{\alpha+\hat{a}} s_{\alpha+\mathbf{r}} s_{\alpha+\mathbf{r}+\hat{a}} \rangle_{3D}, \tag{B.5}$$

for $a \in \{\hat{x}, \hat{y}\}$. In order to increase statistics and enforce the symmetry of the observable, we actually use the average over the correlations between horizontal and vertical pairs, and along two directions $\hat{x}$ and $\hat{y}$ :

$$C(r) = \big(C_x(r\hat{\mathbf{x}}) + C_x(r\hat{\mathbf{y}}) + C_y(r\hat{\mathbf{x}}) + C_y(r\hat{\mathbf{y}})\big)/4. \tag{B.6}$$

The corresponding correlation length $\xi$ is then estimated via

$$\xi = \sqrt{\frac{\langle C(r) r^2 \rangle_r}{\langle C(r) \rangle_r}}. \tag{B.7}$$

This second moment estimator of the correlation length [68] is simpler to numerically estimate in QMC than estimating $\xi$ by fitting the exponential decay of $C(r)$. It is also ignorant about the chosen fitting window, which is often a subtle issue when $\xi$ is estimated by fitting $C(r)$.

Simulations were done at temperature $T = 0.075J$ and the number of imaginary-time slices $M$ was chosen such that $\beta/M = 0.05$.

## C Extrapolation of finite torus energy gaps on the XY transition line

In this appendix we demonstrate how we obtain the CTES of the AT-TFI model on the critical phase transition line from the energy spectrum on finite tori, as shown in Fig. 13 for three different values of $J_{AT}$.

First, for a given $J_{AT} \geq 0$ we tune the strength of the transverse field to the critical point $h_c(J_{AT})$, which was obtained previously from a Binder cumulant analysis using QMC simulations. For this set of parameters, we measure the low-energy spectrum on rotationally symmetric finite size tori with $N \leq 36$ spins. We compute the finite-size energy gaps to the ground state $\Delta_i^N = E_i^N - E_0^N$, where $E_i^N$ denotes the absolute energy of the $i$-th energy level on a cluster of $N$ sites. We then multiply these gaps by the linear system size $L = \sqrt{N}$ to get rid of the dominant scaling of energy gaps at a critical point with dynamical exponent $z = 1$. The eigenstates corresponding to the gaps $\Delta_i^N$ additionally carry quantum numbers $s_i = \pm 1$ under the global spin inversion symmetry $\mathcal{S}$ and an irreducible representation of the square lattice space group [see different symbols in Fig. 13]. We use these quantum numbers to identify matching energy levels among different system sizes.

To obtain the CTES we extrapolate the individual scaled finite-size energy levels $\Delta_i^N \times \sqrt{N}$ linearly in $1/N$ to the thermodynamic limit $N \to \infty$. The so-obtained *gaps*, $\Delta_i \times \sqrt{N}$, together with the quantum numbers of the finite-size energy eigenstates define the CTES of the critical point, as shown in Fig. 5 in the main text.

## D Estimating the critical exponent $\nu$ in the AT-TFI model from QMC simulations

Complementary to the CTES approach used to identify the universality class of the critical points in the AT-TFI model, we apply a standard finite-size scaling approach [44] of the correlation length $\xi$ around the critical points using QMC data [see Fig. 14]. For a given value of $J_{AT} \geq 0$ we plot the scaled correlation length $\xi/L$ against the rescaled distance from the critical point $h_c$ for different linear system sizes $L$. When the proper critical exponents for the analyzed critical point are chosen to rescale the coordinate axes one expects the data to collapse on a single curve for all (large enough) $L$ around $h_c$.

In Fig. 14(a) we show that the correlation length $\xi$ at $J_{AT} = 0$ collapses nicely when the critical exponent for the 3D Ising universality class $\nu_{Ising} = 0.629971(4)$ [51] is used. This is in complete agreement with our analysis in the main text, that the transition of the AT-TFI model with $J_{AT} = 0$ belongs to the 3D Ising$^2$ universality class, where the two sublattices decouple and concurrently undergo a 3D Ising transition, such that $\nu_{Ising^2} = \nu_{Ising}$.

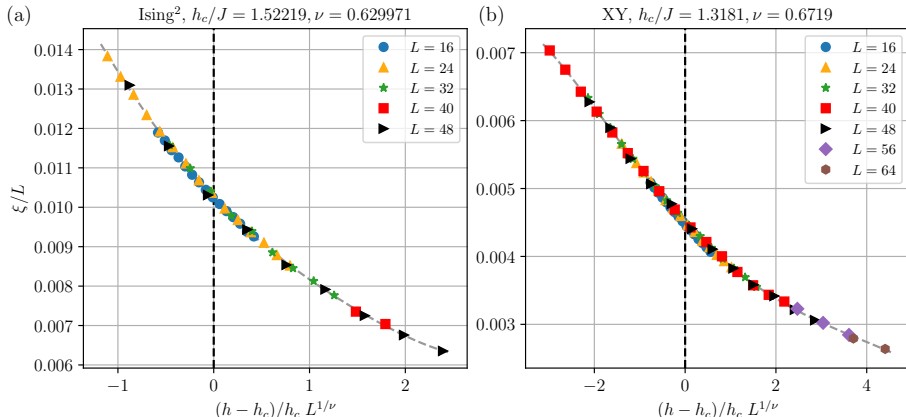

Figure 14: Collapse plots for the correlation length $\xi$ across the Ising$^2$ (a), and the XY transition at $J_{AT}/J = 0.625$ (b). The dashed, grey line shows a polynomial fit of the data as a guide to the eye. The known critical exponent $\nu$ [51] for the Ising (a) and XY transition (b) gives a good collapse of $\xi$.

Figure 14(b) shows the collapse of $\xi$ at $J_{\text{AT}}/J = 0.625$, representative for the XY critical line. We perform the collapse with the critical exponent for the 3D XY universality class $\nu_{\text{XY}} = 0.6719(11)$ [51]. The good data collapse is yet another demonstration that the critical line between the PM and the FM or the $\langle\mu\rangle$ phases in the AT-TFI model belongs to the 3D XY universality class. It is worth to note, that performing the data analysis of $\xi$ with the Ising exponent $\nu_{\text{Ising}}$ visually gives a good collapse, too. This demonstrates the strength of the CTES approach for the identification of critical points, since the CTES is qualitatively different among these universality classes already for systems of only a few ten sites.

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
