# Peer review of "Emergent XY* transition driven by symmetry fractionalization and anyon condensation"

_SciPost Physics, doi:SciPost Phys. 14, 001 (2023)_

## Round 2 · Author Response

We thank the referees for their very positive assessment of our paper and for recommending it for publication in SciPost Physics. We are also grateful for the referees comments, which we will clarify below, point by point.

Response to Anonymous Report 1:

1) We use values of the critical points estimated from QMC simulations of lattices up to 48x48 sites. In particular, we compute the Binder ratio and extract crossing points for different system sizes which are extrapolated to the thermodynamic limit to estimate the value of the critical field h_c/J for a fixed J_{AT}/J. With that, we can estimate the critical points with an accuracy of approximately 3 decimal digits.
We have added a sentence in the manuscript to clarify this.
This accuracy is by far high enough for the subsequent CTES analysis using clusters with a few tens of sites. For (large enough) finite systems, the torus spectrum evolves continuously around the true critical point and will resemble the CTES (which is defined in the thermodynamic limit) for a finite range around the true critical point. This range shrinks with increasing system size, but is larger than the accuracy of the estimates from QMC simulations for the available system sizes in this paper.
In turn, that means that torus spectroscopy can in principle distinguish whether the thermodynamic critical point or a remnant finite-size one is used, when the system sizes become large enough. Torus spectroscopy can then also be used to estimate the value of the critical point. Because of the availability of QMC results, we have however not investigated how precise one could obtain critical points from the torus spectrum in the present manuscript.

2) Trotterization of the transverse field in imaginary time indeed yields a highly anisotropic 3D model, see Eqs.(32, 33) in Appendix B. Therefore, the quantum model H_{AT} considered in this manuscript is microscopically different from the standard classical 3D Ashkin-Teller (CAT) model.
Our argumentation to relate the (2+1)D quantum model to the CAT model is not based on a rigorous mapping of the imaginary time action to the CAT model, but on the similarity of the observed phases in both models for the considered coupling range. In this manuscript we carefully establish the type of phase transitions between these phases in the quantum model using CTES and QMC approaches.
Importantly, the results of this microscopic analysis is fully consistent with the predictions of a phenomenological field theory (see Sec. 2.3), which is only based on the symmetry breaking patterns of the model and the observed phases, but not on the microscopic details.
It is this universal aspect that lets us suggest that the results we find for the types of phase transitions in the quantum model also apply to the classical model, where the symmetry-breaking patterns of the phases are equivalent. In other words, our analysis suggests that the anisotropy of the trotterized model vanishes in the scaling limit around the critical points.
We have added a short paragraph at the beginning of Sec. 3.3 to discuss this point.

3) It is indeed an ubiquitous phenomenon that at a critical point or inside a critical phase where the ultraviolet input at the lattice scale becomes irrelevant due to a diverging correlation length, the system acquires an enlarged emergent (infrared) symmetry in the long wavelength limit, which is a larger symmetry group than the microscopic (ultraviolet) symmetry of the lattice model. For example the deconfined quantum critical points and the Stiefel liquids are both conformally invariant fixed points of such nature. However, the RK point of the quantum dimer model, with a dynamical critical exponent z=2, has very different spatial and temporal correlations, and is not a conformally invariant stable fixed point.
We have added a paragraph at the end of section 4.2 to clarify this point.

4) We indeed think that the torus spectroscopy technique can be a very effective tool to characterize quantum phase transitions on rather small systems.
In previous works [PRL 117, 210401 (2016); PRB 96, 035142 (2017)] we have shown that the CTES of O(N<=3) universality classes computed using ED on clusters of similar sizes as used in this manuscript and extrapolated to the thermodynamic limit (as done here as well) shows good agreement with the CTES obtained directly in the thermodynamic limit with field theoretical methods (i.e. epsilon expansion). 
Furthermore, in the mentioned PRL we have also used QMC methods for systems up to 30x30 sites to extract some levels of the CTES of the Ising universality class. They agree very well with ED calculations on clusters of only up to 40 sites.
On the other hand, our recent study of the more complex chiral Ising GNY universality class [PRB 103, 125128 (2021)] showed that the extrapolation of the energy levels to the thermodynamic limit was not as good from ED data alone as for the O(N) models, in particular for levels at non-zero momenta. The main reason is that in the considered lattice models for the chiral Ising transition the linear light-cone at criticality is much narrower than in the O(n) models considered before. The available momenta closest to the gap-closing momentum on small clusters are, thus, more strongly influenced by warping effects. 
Still, even for those the characteristic, qualitative structure of the CTES was already imprinted in the spectrum of small systems.
It is also worth mentioning that much earlier work on one-dimensional spin chains showed that the critical energy spectrum of small systems can be used to very accurately obtain several critical properties. In particular, a very precise value of the critical point of the J1-J2 model was obtained from exact numerical simulations of chains with only up to 32 sites [Phys. Rev. B 46, 10866 (1992); Phys. Rev. B 54, R9612 (1996)].
To summarize, so far we have observed that the CTES technique works astonishingly well for a large and important class of quantum critical points using only clusters of a few ten sites. 
A very precise estimation of the quantitative values of the CTES gaps and the study of complex, exotic quantum phase transitions might indeed require larger system sizes than the ones used here, but the most important qualitative structure might be already visible on rather small clusters.
Based on our experience we believe that the CTES technique can be very competitive to the more standard technique of measuring critical exponents or, at least, be a strong complementary approach.

Response to Anonymous Report 2:

Eqs. 1, 2?

We have simplified Eqs. (1) and (2) by removing the couplings J_{I_2}’ and J’, which we anyway set to zero for simplicity. In Eq. (2) we now also explicitly write down the translations between the coupling constants among the two models for better readability. We think that using the symbols from Eq. (1) directly in Eq. (2) would make the manuscript harder to understand, since we use “typical" symbols for the interactions (e.g. h = field strength).

Excitation spectrum?

The typically measured critical exponents \eta and \nu are related to the \sigma and \epsilon fields on the sphere. While the values of these levels are rather similar among, e.g., Ising and O(2) (on the torus and on the sphere) additional levels appear for the O(2) model. In particular, an additional low-energy level shows up between the \sigma and \epsilon fields which, on the sphere, is related to a crossover exponent (\eta_c) of the field theory [J. High Energy Phys. 2014, 91 (2014)]).
Thus, the low-energy spectrum directly probes additional universal properties of the universality class, which can be hard to measure in the standard approach. On the sphere they can be directly related to other (subleading) critical exponents, on the torus such a direct mapping is not known. Still the critical spectrum on the torus is highly valuable as it is computationally easily accessible, in stark contrast to the spectrum on the sphere.
The most important feature of the CTES is, however, that the multiplet structure and other quantum numbers of the fields are readily available. This is not the case for the standard critical exponents analysis. This structure is very characteristic and qualitatively distinct among all universality classes considered so far with CTES. In the present manuscript, the emerging two-fold degeneracies of many low-energy levels are a direct consequence of the fact that fields in the O(2) model can have 2-dimensional irreducible representations. This multiplet structure is the most characteristic identifier for the O(2) symmetry class — beyond the precise values of the (scaled) gaps.
We have added a sentence on page 9 where the CTES technique is described to clarify that it is not only the precise values of the CTES levels which are different between universality classes, but in particular the level sequence and their multiplicity structure.

Comment on how the finite-size error depends on the index of the level considered?

We are typically only interested in a few low-energy levels. In particular, we try to obtain the CTES levels which (phenomenologically) can be related to the relevant fields of the critical field theory. For the O(N) models considered so far, we found that the sequence of CTES levels is qualitatively similar to the sequence of scaling dimensions. So, although the CTES does not directly provide access to the scaling dimensions, we only need to track a few low-energy levels (e.g. up to l = 4 for the O(2) CTES shown in Fig. 5). We have observed that these levels all show similar finite-size behaviour.
We have added a sentence in Sec. 2.4. to make the focus on the lowest-energy levels more obvious.

Error bars, comment on estimate of finite-size errors?

Based on our analysis we can only directly estimate an error of the fitting procedure used to extrapolate the finite size values of the energy gaps to the thermodynamic limit. This error is small but does not reflect any systematic errors. In particular, the functional form of the finite-size corrections to the scaling form of the critical energy levels is not known a priori. While the linear extrapolation in 1/N works well for the data in this manuscript, it might be subject to change for larger system sizes or be modified by subleading corrections.
Estimating a justifiable error bar is thus very difficult for the present set of data.
Therefore, and, in particular, since the most important property of the CTES is the qualitative structure of a CTES, we omit giving error bars on the extrapolated values.
We have added a footnote in Sec. 2.4. to comment on why we omit giving error bars for the extrapolated CTES levels in the manuscript.

fig 2 - x-axis related to classical 2D model at zero temperature?

The unconstrained AT-TFI model, Eq. (2), is indeed classical on the x-axis where h=0. It was not the purpose of the present manuscript to study this classical model in detail, but we will list a few of its properties here. In the considered coupling range, for J_{AT}/J < 1 we observe that four states have equally lowest energy. These states are the fully polarized Baxter FM states, consistent with the phase diagram. At J_{AT}/J=1, we then observe that the energy of many other levels becomes identical to the ones of the Baxter FM states and at J_{AT}/J > 1 there is a huge number of ground state levels with the same energy. While we have not investigated in detail if the zero temperature entropy density is indeed macroscopically large, our results indicate that an order-by-disorder mechanism stabilizes the <mu> phase for arbitrarily small transverse field. 
We have not investigated the classical phase diagram in more detail, in particular for larger absolute values of |J_{AT}/J| > 1.
We have added a sentence in the manuscript to mention the large classical ground state degeneracy and the prospective order-by-disorder mechanism.

p11 (weakly) first order transition?

Fig. 3b shows a cut through the first order phase transition line relatively far away from its termination point at J_{AT}=0. There phase coexistence is clearly visible. Closer to the termination point, however, the signal of phase coexistence becomes less distinct for the available system sizes. In fact, according to the phenomenlogical quantum field theory for our model, the first order transition can be made arbitrarily weak when J_{AT} approaches J_{AT}=0 from below.
We have adapted the sentence in the manuscript to clarify this point.

Emergent XY* symmetry?

As we point out in the manuscript, we attribute the small (finite-size) splitting of the Sz=2 levels and their sequence in the CTES (see Fig. 5) to the sign of the dangerously irrelevant 4-fold anisotropy term along the XY transition line. The size of the effective coupling constant, however, cannot be accessed from the CTES. Still, this analysis nicely demonstrates that a CTES (and its finite-size scaling properties) can provide access to intricate properties of the corresponding field theory, at least qualitatively.

Sec 4.1 - Correspondence between sigma_x and phi_A phi_B?

We have added a sentence at the beginning of section 4.1. which repeats the mapping between \sigma and \mu operators in the two considered models. The fields phi_1, phi_2 defined in the phenomenological field theory description, Eq. 11, are coarse-grained versions of the \mu operators on the two sublattices A and B on the square lattice.

p20 “it is impossible to satisfy the algebra” can the meaning in this language of the breaking of Rz be explained?

We have added two sentences on page 20 to clarify why it is impossible to break the Ising symmetry while preserving both translation and C4 rotation.

Repetition?

We have removed this sentence and shortened section 4.3.2 to remove unnecessary repetition. For the clarity and accessibility to a broad audience, we refrain from substantially shortening the rest of the manuscript.

References "...*" transitions?

We have added the references [4, 11] again on page 3, where the Ising* transition is first mentioned.

footnote 3, typos

We have included the footnote in the main text and have corrected the typos.

---

## Round 2 · List of Changes

• We have added references [4, 11] in the introduction where the starred phase transitions are introduced

  • We have added reference [26] on several positions, in which the (Ising^2)* transition is already discussed

  • We have simplified Eq. (1) and the subsequent paragraph by removing the unnecessary coupling J_{I_2}'

  • We have adapted Eq. (2) by removing the unnecessary coupling J' and adding the translations between coupling of the two different models directly here. Accordingly, we have adapted the subsequent paragraph

  • Figure 1: We have removed the couplings J_{I_2}' and J' and adjusted the caption accordingly.

  • Footnote 3 was included in the main text

  • We have added a sentence in Sec. 2.4 to clarify how the phase transition points are estimated from QMC

  • We have added a few sentences in Sec. 2.4 to clarify the qualitative structure of the CTES and which levels we track there

  • We have added a footnote to comment on why we omit giving error bars for the extrapolated CTES levels

  • We have added a short paragraph at the end of Sec. 3.1. discussing the potential order-by-disorder origin of the \mu phase

  • At the beginning of Sec. 3.2 we have added a sentence clarifying the "weakness" of the first order transition.

  • Fig. 5: We have added the multiplicities of the individual CTES levels on the left and the right of the figure and adapted the caption accordingly

  • We have added a short paragraph about the anisotropic classical model after trotterization and the 3D CAT model at the beginning of Sec. 3.3

  • We have added a sentence about the relation between \sigma and \mu operators at the beginning of Sec. 4.1

  • In Sec. 4.2. we have adapted the sentence describing the symmetry breaking restrictions to make it better understandable

  • We have added a paragraph at the end of Sec. 4.2 to discuss the implications of the non-trivial fractionalization class on the emergent symmetries of the phase transition

  • We have shortened paragraph 4.3.2 to remove unnecessary repetition

  • We have fixed several typos throughout the manuscript

  • We have updated the affiliations of AML

  • Appendix A: We have added two paragraphs discussing the J_{I_2}' and J' couplings, which have been removed from the main text

---

## Editorial Decision

published